# Exploiting interconnected synthetic lethal interactions between PARP inhibition and cancer cell reversible senescence

Hubert Fleury[1,2,6], Nicolas Malaquin[1,2,6], Véronique Tu [1,2], Sophie Gilbert[1,2], Aurélie Martinez[1,2], Marc-Alexandre Olivier [1,2], Skye Alexandre Sauriol [1,2], Laudine Communal[1,2], Kim Leclerc-Desaulniers[1,2], Euridice Carmona[1,2], Diane Provencher[1,2,3], Anne-Marie Mes-Masson[1,2,4] & Francis Rodier[1,2,5]

Senescence is a tumor suppression mechanism defined by stable proliferation arrest. Here we demonstrate that the known synthetic lethal interaction between poly(ADP-ribose) polymerase 1 inhibitors (PARPi) and DNA repair triggers p53-independent ovarian cancer cell senescence defined by senescence-associated phenotypic hallmarks including DNA-SCARS, inflammatory secretome, Bcl-XL-mediated apoptosis resistance, and proliferation restriction via Chk2 and p21 (CDKN1A). The concept of senescence as irreversible remains controversial and here we show that PARPi-senescent cells re-initiate proliferation upon drug withdrawal, potentially explaining the requirement for sustained PARPi therapy in the clinic. Importantly, PARPi-induced senescence renders ovarian and breast cancer cells transiently susceptible to second-phase synthetic lethal approaches targeting the senescence state using senolytic drugs. The combination of PARPi and a senolytic is effective in preclinical models of ovarian and breast cancer suggesting that coupling these synthetic lethalities provides a rational approach to their clinical use and may together be more effective in limiting resistance.

[1] Centre de recherche du Centre hospitalier de l'Université de Montréal (CRCHUM), Montreal H2X 0A9 QC, Canada. [2] Institut du cancer de Montréal, Montreal H2X 0A9 QC, Canada. [3] Division of Gynecologic Oncology, Université de Montréal, Montreal H3C 3J7 QC, Canada. [4] Department of Medicine, Université de Montréal, Montreal H3C 3J7 QC, Canada. [5] Department of Radiology, Radio-Oncology and Nuclear Medicine, Université de Montréal, Montreal H3C 3J7 QC, Canada. [6]These authors contributed equally: Hubert Fleury, Nicolas Malaquin. Correspondence and requests for materials should be addressed to A.-M.M-M. (email: anne-marie.mes-masson@umontreal.ca) or to F.R. (email: rodierf@mac.com)

Poly(ADP-ribose) polymerase 1 (PARP1) plays an important role in DNA damage repair and PARP inhibitors (PARPi) have been explored as anticancer agents based on their ability to induce synthetic lethality in the context of homologous recombination (HR) deficiencies, such as those caused by BRCA1/2 mutations[1,2]. PARPi are used as maintenance therapy for the treatment of high-grade serous epithelial ovarian cancer (HGSOC)[3]; however, clinical trials have revealed varied responses in the context of either HR deficiency or proficiency[3], suggesting that HR-independent modes of action are involved in the observed clinical effects of PARPi[3,4]. Treatment resistance associated with chronic maintenance therapy is challenging, and although the emergence of mutations restoring BRCA function explain some occurrence[5], other resistance mechanisms remain unknown. Overall, new combination therapies are being tested in the clinic to improve PARPi efficacy[5,6].

PARPi induce DNA damage accumulation[7], leading to DNA damage responses (DDR) that favor cell cycle checkpoints, and tissue remodeling. The DDR orchestrates cell fate decisions, such as transient or prolonged proliferation arrest (senescence), apoptosis (cell death), or mitotic catastrophe[8]. In normal cells, senescence is considered a state of irreversible proliferation arrest maintained by the activity of the p53/p21 and p16/Rb pathways[8,9]. However, classic pathways of senescence-associated (SA) proliferation arrest (SAPA) are almost always mutated in cancer[10]. Whether therapy-induced senescence (TIS) in cancer cells is irreversible thus remains unclear, particularly with emerging evidence demonstrating that rare cancer cells within an otherwise senescent population may escape senescence[9,11–14]. Senescent cells are an integral component of tissue repair programs via their inflammation-regulating SA secretory phenotype (SASP) that modulates interactions with surrounding epithelial, stromal, immune, and stem cells[8,15,16]. In the context of TIS, senescence generates context-dependent beneficial or detrimental effects, sometimes leading to improved treatment efficacy via proliferation arrest, or by contrast, promoting long-term cancer recurrence through the creation of a protective niche for surviving cancer stem cells[8,10,17]. Given the major cell autonomous and nonautonomous consequences of TIS, as well as the potential to escape from this cell fate, TIS has major implications for cancer treatment resistance and recurrence. Therefore, TIS has become a target for combination-therapy enhancements via modulation of SA phenotypes[18–21].

To explore the cellular mechanisms of action of PARPi we focus on cell fate decisions that we observed in HGSOC cell lines reflecting a PARPi sensitivity spectrum[4]. We found that PARPi induces TIS and all major senescence hallmarks in HGSOC cells. PARPi similarly induces TIS in breast cancer cells, and both cancer cell types are sensitive to senolytic drugs, which convert senescence to apoptosis, inducing a synergistic lethal effect[18,20,21]. Using preclinical cancer xenograft models, we demonstrate that an effective PARPi-senolytic combination-therapy for ovarian and breast cancer reduces tumor burden. Unlike previously described stable SAPA, we show that PARPi-induced TIS is reversible, revealing an unstable senescence-like state. Importantly, we demonstrate that reversibly senescent cells induced by cancer therapy provide a treatment window for opportunistic elimination using synergistic senolytic drugs. Our data suggest that the clinical application of PARPi as maintenance therapy may favor recurrence and resistance by inducing a pharmacologically targetable inflammation-regulating reversible senescence-like state.

## Results

### PARPi-treated HGSOC cells develop a senescence-like phenotype.

Previously, we investigated the sensitivity of 18 HGSOC cell lines to the PARPi Olaparib, using clonogenic assays[4]. To determine the process by which the cells lose proliferation potential, we performed live-cell imaging assays to evaluate real-time cell fate decisions in four of these cell lines, OV1369(R2), OV90, OV4453, and OV1946. In accordance with the molecular characteristics of the disease[22], these HGSOC cell lines carry TP53 mutations and have high rates of copy number anomalies[23–26]. In particular, OV4453 carries a BRCA2 mutation that is likely responsible for PARPi sensitivity[4,23]. Real-time imaging confirmed dose-dependent Olaparib-mediated inhibition of cell proliferation in which higher concentrations were required for two cell lines and $IC_{50}$ were consistent with those obtained using clonogenic assays (Fig. 1a, Supplementary Fig. 1A). Interestingly, live-cell imaging revealed that inhibition of cell proliferation was not accompanied by significant cell detachment. This was confirmed by correspondingly small increases in total cumulative cell death/apoptosis, as only 20–40% of cells were cumulatively AnnexinV and/or DRAQ7 positive 6 days after treatment initiation, even at the highest Olaparib concentrations (Fig. 1b, Supplementary Fig. 1B). However, real-time images revealed treatment-associated changes in cell morphology, including cell enlargement that started at day 3 and became more pronounced at day 6 (Supplementary Fig. 1C), suggesting a senescence cell fate response.

Senescent cells including TIS cells are characterized by a series of SA phenotypes displayed in vitro as well as in vivo[8,10], and several of these features are initiated by DNA damage and the DDR[27], corresponding to events triggered by PARPi[1,2,4]. In addition to SAPA and morphological changes, senescent cells increase lysosomal mass promoting SA-β-galactosidase (SAβgal) activity and undergo chromatin remodeling that accompanies gene expression profiles favoring SA apoptotic resistance (SAAR) and the development of a SASP secretome[15,28,29]. We observed a concentration-dependent increase in SAβgal positive cells in all cell lines at day 6 after Olaparib treatment, with higher levels of senescence detected in cell lines with high PARPi $IC_{50}$. (Fig. 1c, Supplementary Fig. 1D), consistent with their lower levels of cumulative apoptosis (Fig. 1b). We also confirmed cell enlargement using flow cytometry analysis of cell granularity (SSC) and size (FSC) observing two distinct cell populations, R1 (normal size) and R2 (enlarged cells) (Fig. 1d, Supplementary Fig. 1E), whereas R2/R1 ratios were significantly increased in Olaparib-treated cells at day 6 (Fig. 1d). In addition, time-dependent higher levels of secreted interleukins IL-6 and IL-8 suggested a SASP (Fig. 1e, f), which was confirmed by multiplex analysis of SASP components (Supplementary Fig. 1F). Persistent DNA damage is one of the major causes of cellular senescence[27]. Accordingly, we observed increased levels of γH2AX and 53BP1 indicating persistent nuclear DNA damage foci after Olaparib exposure (Fig. 1g, Supplementary Fig. 1G, H). Live-cell imaging showed apparent SAPA, a key hallmark of senescence[8], which we confirmed with short EdU pulse-labeling assays that revealed decreased DNA synthesis in Olaparib-treated HGSOC cells at 6 days (Fig. 1h, Supplementary Fig. 2A). However, unlike a canonical senescence response, cell proliferation was not completely arrested, as three out of four HGSOC cell lines incorporated EdU following longer pulses (Fig. 1i), indicating slow cycling cells. This was confirmed by a cell cycle analysis at 6 days post-treatment showing accumulation at the G2/M phases of the cell cycle and appearance of cell populations presenting DNA content beyond 4N (Fig. 1j, Supplementary Fig. 2B), consistent with described effects of Olaparib on the cell cycle[30]. Thus, PARPi induced a senescent-like state with stalled cell cycle progression in HGSOC cells. Apart from Olaparib, additional PARPis have been evaluated in clinical trials for the treatment of HGSOC patients including Niraparib and Talazoparib, which possess distinct PARP1-DNA trapping activities[2,31,32].

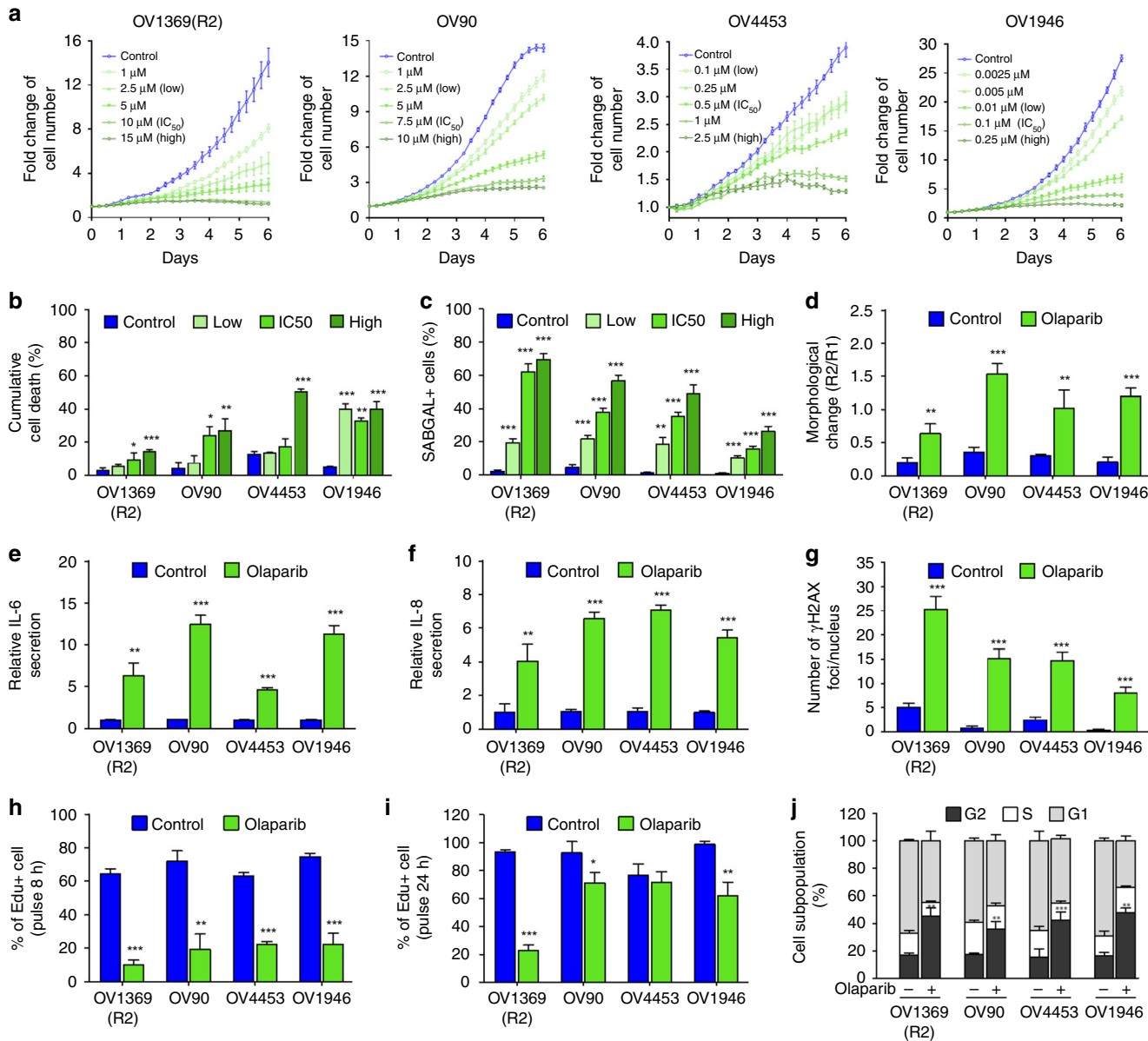

**Fig. 1** Olaparib induces a senescence-like phenotype in HGSOC cell lines. **a** Cell proliferation curves of HGSOC H2B-GFP cell lines exposed to increasing concentrations of Olaparib. **b, c** HGSOC dead cells analyzed by flow cytometry (**b**) and SAβgal positive HGSOC cells (**c**) following 6 days treatment with selected Olaparib concentrations (Supplementary Fig. 1A). **d** HGSOC cell morphology analyzed by flow cytometry following 6 days of treatment with Olaparib IC$_{50}$ concentrations (see Supplementary Fig. 1A, E for details). **e, f** Levels of IL-6 (**e**), IL-8 (**f**) were measured by ELISA assay following 6 days treatment with Olaparib IC$_{50}$ concentrations. **g** Number of γ-H2AX foci per nucleus in HGSOC cells lines following 6 days of treatment with Olaparib IC$_{50}$ concentrations. **h, i** Analysis of 8-h (**h**) or 24-h (**i**) EdU pulse after 6 days exposure of HGSOC cells to Olaparib IC$_{50}$ concentrations. **j** Flow cytometry analysis of cell cycle populations following 6 days exposure of HGSOC cells to Olaparib IC$_{50}$ concentrations. Data in (**a**) are representative curves of at least three independent experiments. For all the data, the mean ± SEM of three independent experiments is shown. Data were analyzed using the two-tail Student *t* test. *Denotes $p < 0.05$, **$p < 0.01$, and ***$p < 0.001$

Dose–response curves for OV1369(R2) cells showed concentration-dependent inhibition of cell proliferation for both Niraparib and Talazoparib including induction of a senescence-like phenotype (Supplementary Fig. 3A–C), showing that induction of TIS is not restricted to Olaparib, but is a common response to PARPi.

**PARPi-induced senescence is mediated by p21 and Chk2.** In response to DNA damage, cyclin-dependent kinase inhibitors (CDKi) regulate the cell cycle. To elucidate mechanisms behind cell cycle checkpoints induced by Olaparib in HGSOC cells, we

profiled key CDKi by using gene expression analysis and observed that p21(*CDKN1A*) was significantly upregulated in all four HGSOC cell lines, whereas p16 (*CDKN2A*) was either undetected or did not increase (Fig. 2a). SAPA usually relies heavily on the p53/p21 and p16/Rb pathways[9], but HGSOC cells are defined by a *TP53* mutant status[22], which was confirmed for HGSOC cells in this study[23–26]. Therefore, increased levels of the direct p53 transcriptional target p21 are unexpected. However, p53-independent activation of p21 has been reported during embryonic- and oncogene-induced senescence[33] and following overexpression of the Chk2 DDR kinase in epithelial cancer cells[34]. To test whether a Chk2-p21 pathway similarly regulates

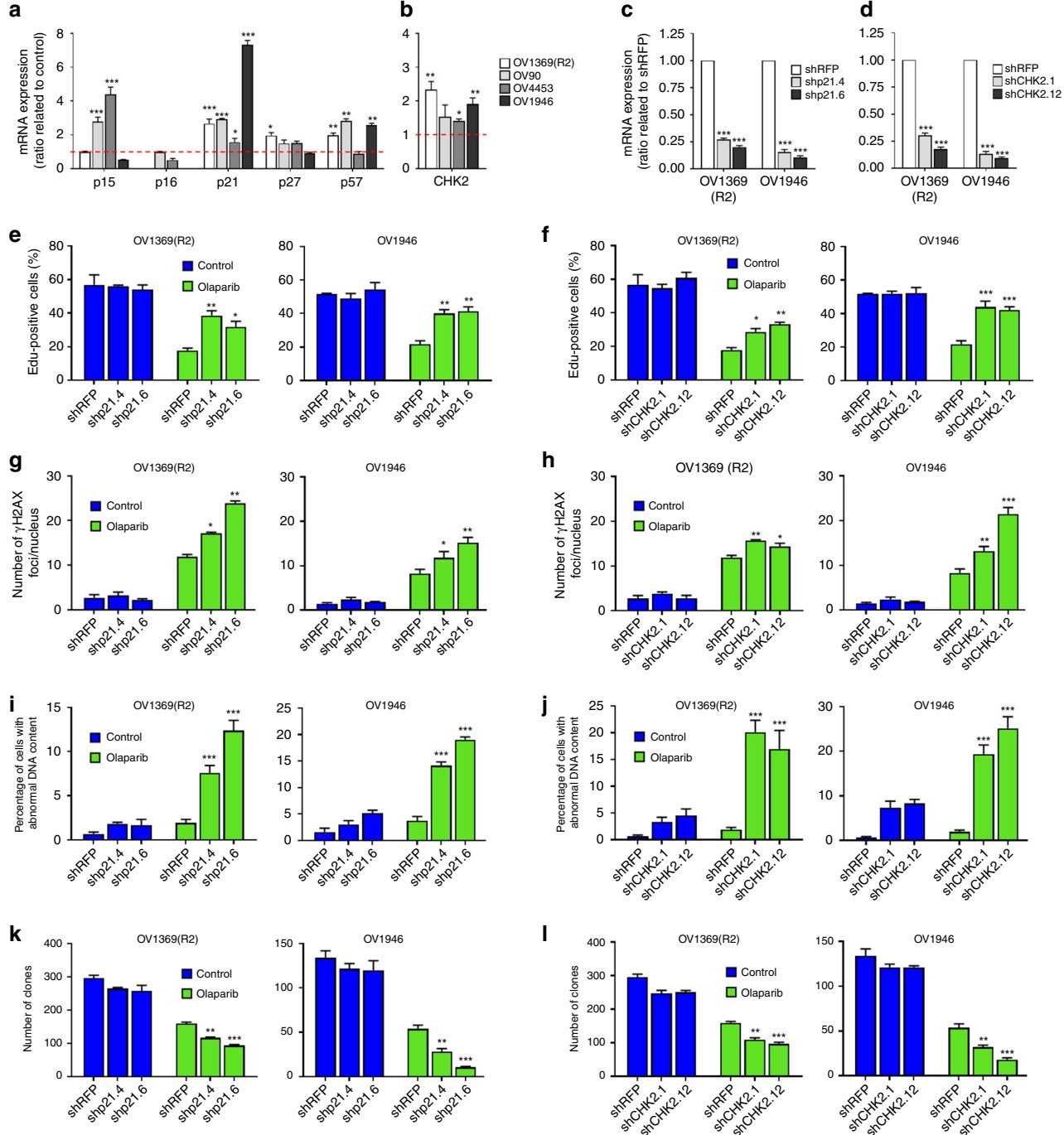

**Fig. 2** Involvement of p21 and Chk2 in Olaparib-induced senescence. **a, b** Relative mRNA levels of p15, p16, p21, p27, and p57 (**a**) or Chk2 (**b**) evaluated by real-time Q-PCR in HGSOC cells treated with Olaparib IC$_{50}$ concentrations for 6 days. The values represent the fold change expression related to nontreated controls. **c, d** Real-time Q-PCR showing gene silencing by shRNA against p21 (**c**) or Chk2 (**d**) in HGSOC cells. **e, f** Analysis of 8-h EdU pulse after 6 days exposure of shp21 (**e**) or shChk2 (**f**) infected HGSOC cells to Olaparib IC$_{50}$ concentrations. **g, h** Number of γ-H2AX foci per nucleus in shp21 (**g**) or shChk2 (**h**) infected HGSOC cells treated with Olaparib IC$_{50}$ concentrations for 6 days were determined by analyzing >150 cells per condition. **i–j** Flow cytometry analysis of DNA content following 6 days exposure of shp21 (**i**) or shChk2 (**j**) infected HGSOC cells to Olaparib IC$_{50}$ concentrations. **k, l** Clonogenic assays were performed on shp21 (**k**) or shChk2 (**l**) infected HGSOC cells treated or not with Olaparib IC$_{50}$ concentrations for 6 days. For all the data, the mean ± SEM of three independent experiments is shown. Data were analyzed using the two-tail Student t test. *Denotes $p < 0.05$, **$p < 0.01$, and ***$p < 0.001$

PARPi-induced proliferation arrest in HGSOC cells, we verified the Chk2 (*CHK2*) expression by quantitative polymerase chain reaction (Q-PCR) in cells treated with Olaparib for 6 days and observed significant upregulation in three of the four cell lines (Fig. 2b). We then performed p21 (Fig. 2c and Supplementary

Fig. 4A) or Chk2 (Fig. 2d and Supplementary Fig. 4A, B) depletions in OV1369(R2) and OV1946 cell lines using lentivirus-delivered stable shRNAs. Depleting either p21 or Chk2 partially prevented the proliferation arrest of Olaparib-treated HGSOC cells (Fig. 2e, f, Supplementary Fig. 4C, D) leading to an increase

in γH2AX foci number (Fig. 2g, h and Supplementary Fig. 4E), an increase in abnormal DNA content (Fig. 2i, j and Supplementary Fig. 4D) and an increase in the micronuclei number (Supplementary Fig. 4E, F), providing strong evidence for genomic instability and mitotic catastrophe. Finally depleting either p21 or Chk2 redirected Olaparib-treated cells from senescence toward cell death, as observed by significantly decreased colony formation (Fig. 2k, l). In addition, except for OV1369(R2) treated by shp21, significantly lower cell number ratio and higher cell death was observed (Supplementary Fig. 5A–C). These results suggest that a Chk2-p21 DDR network is responsible for mediating a cell cycle arrest that favors survival over mitotic catastrophe and allows other SA phenotypes.

**The PARPi-induced senescence in HGSOC cells is reversible.** Although senescence is defined as a terminal arrest of cell division, rare cells within otherwise chemo- or radiation-induced senescent populations have been suggested to escape senescence supporting a potential mechanism of cancer treatment resistance[11,12,17]. Therefore, we investigated whether the senescence-like state induced by Olaparib in p53-mutant HGSOC cells was reversible. Cells continuously treated with Olaparib displayed a senescent-like phenotype with DDR, proliferation arrest and other SA hallmarks such as SAβgal and SASP on day 6 (Fig. 1b–j, Supplementary Figs. 1B–H, 2A, B). To test the reversibility of SA hallmarks, media was changed to drug-free conditions at either day 3 (early SA hallmarks) or day 6 (senescence). Cells responded with a recovery of proliferation potential (3 days, 4 out of 4; 6 days, 3 out of 4), and DNA synthesis (3 days 4 out of 4; 6 days 4 out of 4) (Fig. 3a–e). To further verify whether senescence recovery occurred in rare isolated cells or in the majority of cells within senescent populations, individual cells were evaluated by clonogenic assays that were performed during and after Olaparib treatment (drug release). The majority of cells within treated populations recovered proliferation capacity after a 6 days treatment, where significant numbers of colonies were formed (Fig. 3f), although this was more pronounced in Olaparib IC50-high OV1369(R2) and OV90 cell lines. We postulate that recovery in the Olaparib IC50-low OV4453 and OV1946 cell lines is delayed due to their higher levels of DNA repair defects[4], resulting in more prolonged DNA damage reflected in the lower recovery rates observed with colony assays (Fig. 3f). Thus, the reversible senescence-like phenotype induced by Olaparib is observed in a large proportion of HGSOC TIS cells within the senescent cell population, and these cells may directly contribute to rapid therapy resistance and/or recurrence.

**Bcl2-family inhibitors synergize with PARPi in HGSOC cells.** A key feature of senescent cells is apoptosis resistance (SAAR)[35], which involves the increased expression of anti-apoptotic Bcl2 family genes[18,20]. SAAR is a targetable hallmark of senescence and Bcl2-family inhibitors can redirect senescent cells toward apoptosis[18,20,21]. To explore the possibility that a combination therapy of PARPi and Bcl2-family inhibitors could kill HGSOC via TIS we first used ABT-263, a Bcl2/Bcl-XL inhibitor extensively used in clinical trials and the first to be identified as a senolytic[36]. To highlight potentially additive or synergistic effects, we used Olaparib IC50 concentrations (Supplementary Fig. 1A) in combination with selected ABT-263 concentrations that had none or minimal effects as single agent in untreated HGSOC cell line (i.e., 0.25 μM for OV1369(R2), and 2.5 μM for the other HGSOC cell lines; all ABT-263 sensitivity data are presented in Supplementary Fig. 6A, C). We observed a significant potentiation of cell killing as measured in proliferation assays and total cumulative cell death (Fig. 4a–d). To evaluate the therapeutic potential of this

combination, we calculated combination indexes (CI) using a range of ABT-263/Olaparib concentrations at fixed ratios (Supplementary Fig. 6B). CI value below 0.9 in all the concentration ranges used with all four cell lines strongly supported synergistic activity (Fig. 4e, f).

Because ABT-263 targets both Bcl-2 and Bcl-XL proteins, we first evaluated the contributions of these two Bcl2-family members in HGSOC. Using the TCGA mRNA expression dataset[22] we found that Bcl-XL expression is significantly higher than that of Bcl2 in HGSOC samples (Fig. 4g). Using our HGSOC cell lines we confirmed this result by Q-PCR and western blot analyses (Fig. 4h, i). Subsequently, we showed that Olaparib-treated cells have higher Bcl-XL levels than control nontreated ones (Fig. 4i), indicating a SAAR state. We further dissected the contributions of these two Bcl2-family members using specific inhibitors for Bcl2 (ABT-199, Venetoclax) or Bcl-XL (A-1155463 and A-1331852), which can also act as context-dependent senolytic agents[37]. We analyzed combination treatments mixing IC50 concentrations of Olaparib (Supplementary Fig. 1A) with varying concentrations of Bcl2-family inhibitors (Supplementary Fig. 6D) using the Bliss independent model[38], in which negative values indicate antagonism, values around zero indicate additive effects and positive values indicate synergy. Overall, synergistic killing by co-treatment of Olaparib with ABT-263 or Bcl-XL inhibitors (A-1155463 and A-1331852) was observed in both cell lines, but not when combining specific Bcl2 inhibitor (ABT-199) to PARPi (Fig. 4j, Supplementary Fig. 7). Again, similarly to Olaparib, Niraparib, and Talazoparib treated cells have higher Bcl-XL and IL-8 (Supplementary Fig. 8A) and can be synergistically killed when combined with the senolytic ABT-263 (Supplementary Fig. 8B, C). These results confirm that HGSOC-TIS cells are particularly sensitive to Bcl-XL inhibition. Interestingly, Bliss scores for the specific Bcl-XL inhibitors (A-1155463 and A-1331852) were slightly higher in the Olaparib IC50-high OV1369(R2) cell line than the IC50-low OV1946 (Fig. 4j), supporting the idea that higher levels of senescence in PARPi-treated Olaparib IC50-high cells coincide with more senolytic potential.

**Synergy between PARPi and senolytic occurs in the context of senescence.** To confirm that Bcl2/Bcl-XL inhibitors were preferentially targeting senescent-like HGSOC cells, we performed senolytic-Olaparib combination-treatments at time points that either preceded or were concomitant with the development of senescence-like phenotypes as these are known to mature over several days[18,20]. In our conditions, we observed a gradual senescence-like phenotype induction starting 3 days after Olaparib treatment (Supplementary Fig. 9A–D). When both Olaparib and ABT-263 were incubated together for only three days, prior to senescence-like phenotypes maturation, only a mild increase in cell death was observed (Fig. 5a, b). In contrast, the combination of senolytics either three or six days after Olaparib exposure, when senescence-like phenotypes were readily detectable, inhibited overall proliferation and induced apoptosis in a manner comparable to the continuous combination-treatment from day one (Fig. 5c, d, Supplementary Fig. 9E). Importantly, after senescence reversal, when HGSOC cells no longer harbored senescent-like phenotypes, ABT-263 sensitivity was lost, particularly in the OV1369(R2) and OV90 that recovered normal proliferation more readily (Fig. 5e, f). Indeed, levels of γH2AX and 53BP1 nuclear foci (Supplementary Fig. 9F–H) or secreted IL-6 and IL-8 (Supplementary Fig. 9I, J) after release reverted to the same levels as matched controls for the OV1369(R2) and OV90 cell lines. However, this effect was less pronounced for the OV1946 cells and even sometimes reversed for the OV4453, consistent with the slower growth recovery curves for these cells (Fig. 3a–e).

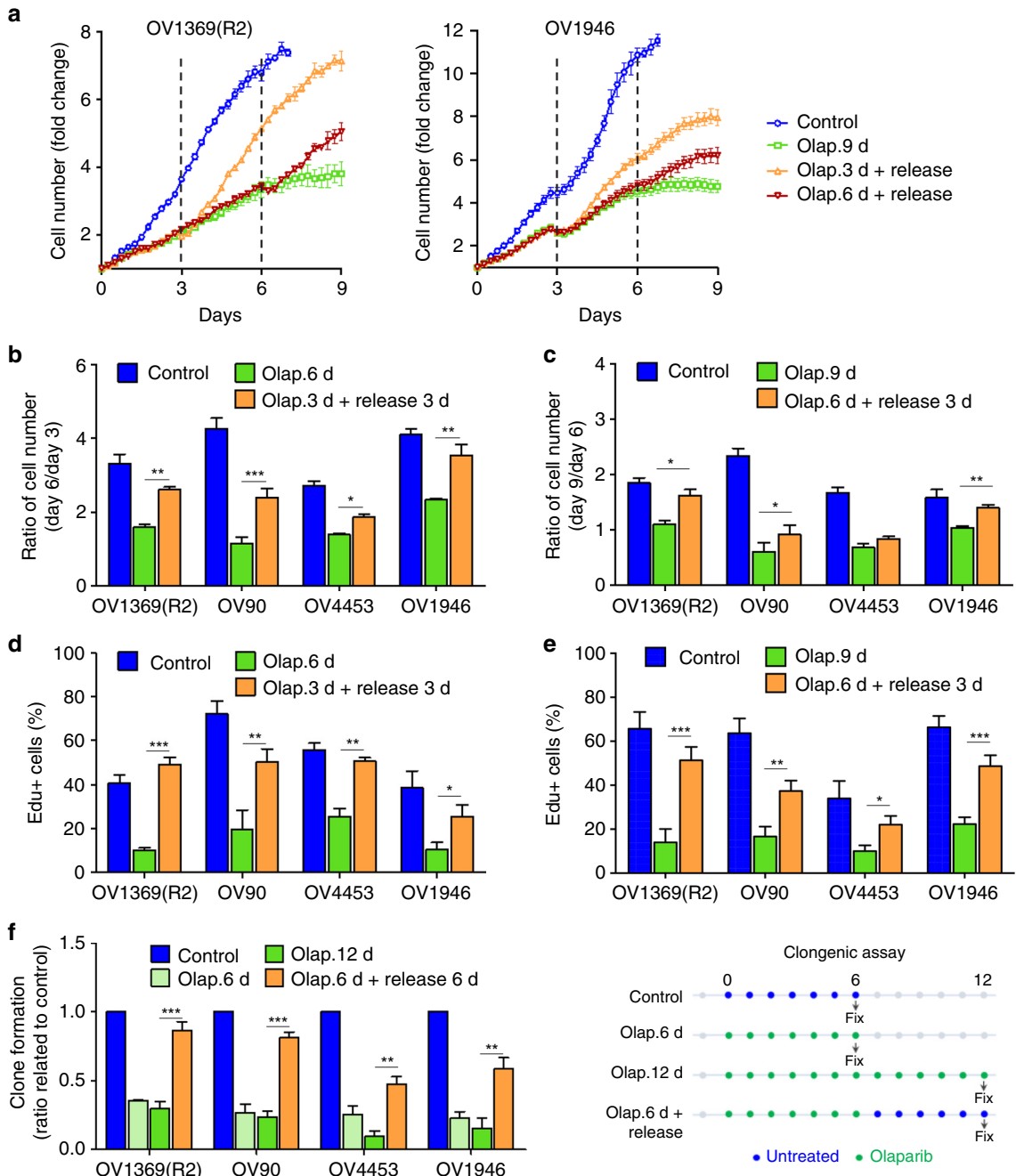

**Fig. 3** Senescent phenotype induced by Olaparib is reversible. **a** Representative proliferation curves of HGSOC H2B-GFP cells treated with Olaparib $IC_{50}$ concentrations for 9 days (9 d), or treated for 3 and 6 days (3 and 6 d) then without for the remaining days (Release). **b–e** Fold change in cell number (**b, c**) and analysis of 8-h EdU pulse (**d, e**) for HGSOC H2B-GFP cells treated with Olaparib $IC_{50}$ for 6 days or 3 days then without for another 3 days (**b, d**); or treated with Olaparib $IC_{50}$ for 9 days or 6 days then without for another 3 days (**c, e**). **f** Clonogenic assays were performed on untreated HGSOC cells, cells treated with Olaparib for 6 or 12 days, or treated with Olaparib for 6 days then without for the next 6 days. Data in (**a**) are representative curves of at least three independent experiments. For all the data, the mean ± SEM of three independent experiments is shown. Data were analyzed using the two-tail Student $t$ test. *Denotes $p < 0.05$, **$p < 0.01$, and ***$p < 0.001$

In addition to the upregulation of Bcl2 family members, other anti-apoptotic pathways such as PI3K/Akt, p53/p21/serpines, receptor tyrosine kinases, and HIF-1α, are activated in senescent cells providing additional critical vulnerable opportunities for senolytics[39]. To further confirm that senolytics in general synergize with the senescent-like state induced by Olaparib in HGSOC cells, we tested a senolytic panel consisting of: Dasatinib (D) (Src tyrosine kinase inhibitor), Quercetin (Q) (bioflavonoid

with a broad spectrum of action targeting several pathways, including PI3K/Akt, p53/p21/serpines and Bcl2/Bcl-XL), and two derivatives of Q, Fisetin (F) (PI3K/Akt inhibitor) and Piperlongumine (PPL) (targeting p53/p21/serpines and PUMA)[39]. We determined senolytics $IC_{50}$ for OV1369(R2) and OV1946 cells (Supplementary Fig. 10A), and Bliss scores were calculated for combination treatments between Olaparib ($IC_{50}$ concentration, Supplementary Fig. 1A) and each of the senolytic

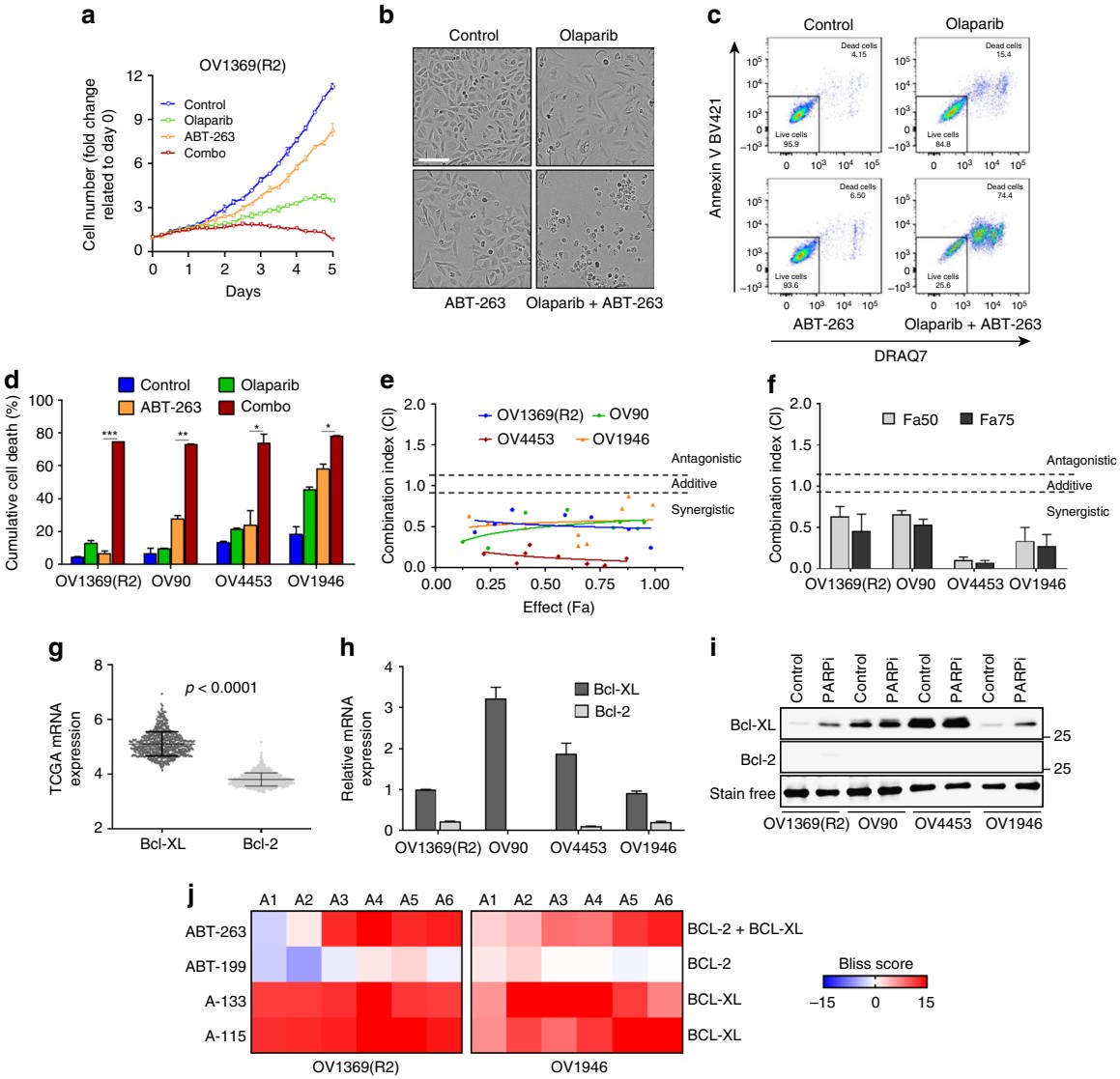

**Fig. 4** Olaparib and Bcl2/Bcl-XL inhibitors are synergistic. **a** Cell proliferation curves of OV1369(R2) H2B-GFP cell line treated with Olaparib (10 μM), ABT-263 (0.25 μM), or Olaparib/ABT-263 (10/0.25 μM) for 6 days. **b, c** OV1369(R2) treated for 6 days with Olaparib (10 μM), ABT-263 (0.25 μM) or the combination Olaparib/ABT-263 (10/0.25 μM). Representative images of cell morphology and cell fate. Scale bar, 150 μm (**b**). Representative flow cytometry analysis of apoptosis showing DRAQ7 positive cell population (X-axis) and AnnexinV positive cells (Y-axis) (**c**). **d** Percentage of dead cells for all four HGSOC cell lines treated with their respective PARPi $IC_{50}$ combined to ABT-263 (0.25 μM for OV1369(R2) and 2.5 μM for the other cell lines) analyzed by flow cytometry as in (**c**). **e, f** CI values for the entire fraction affected (Fa) (**e**) and the mean ± SEM of the CI at Fa 0.50/0.75 (**f**) of HGSOC cells expressing H2B-GFP for the combined treatment (Olaparib/ABT-263) at specific drug ratios (Supplementary Fig. 6B). **g** Comparison of BCL2 and BCLXL mRNA expression in HGSOC according to the TCGA dataset (530 samples, cBioPortal). **h** Relative mRNA expression of BCL2 and BCLXL evaluated by real-time Q-PCR in HGSOC cell lines. **i** Western blot analyses of Bcl-XL and Bcl2 proteins in HGSOC cells treated with Olaparib for 6 days. **j** Heat map of Bliss scores for combination treatments between Olaparib $IC_{50}$ and different concentrations of ABT-263, ABT-199, A133, or A115 (see Supplementary Fig. 6D) of HGSOC cells expressing H2B-GFP. Data in (**a–c, e, i**) are a representation of at least three independent experiments. For all the data, the mean ± SEM of three independent experiments is shown. Data were analyzed using the two-tail Student t test. * Denotes $p < 0.05$, **$p < 0.01$, and ***$p < 0.001$

drugs (several concentrations according to their $IC_{50}$ values, Supplementary Fig. 10B). We found that synergistic effects (positive Bliss scores) between Olaparib and tested senolytic drugs were consistent in both cell lines and predominantly occurred when combination treatments were performed on days 3–6 when compared to days 0–3 (Fig. 5g, Supplementary Fig. 10C), indicating selective targeting of the senescent-like state. Briefly, OV1369(R2) and OV1946 cells showed higher Bliss scores for ABT-263, A115463 (A-115), and PPL (Fig. 5g). To complement the Bliss score, we calculated a senolytic index (SI) based on the ratios of surviving cells (over control) for each combination in order to reflect the capacity of senolytic drugs to

specifically eliminate senescent cells. The SI reached the same overall conclusion but additionally highlighted the enhanced senolytic effect of the combination in OV1369(R2) cell line (Fig. 5h). Taken together, these results suggest that multiple anti-apoptotic pathways play a critical survival role in HGSOC TIS cells induced by Olaparib.

**Targetable PARPi-induced senescence occurs in breast cancer cells**. In addition to HGSOC, PARPis are highly relevant to breast cancer, owing to the relatively frequent BRCA mutations in this type of malignancy[40]. Olaparib dose–response curves for

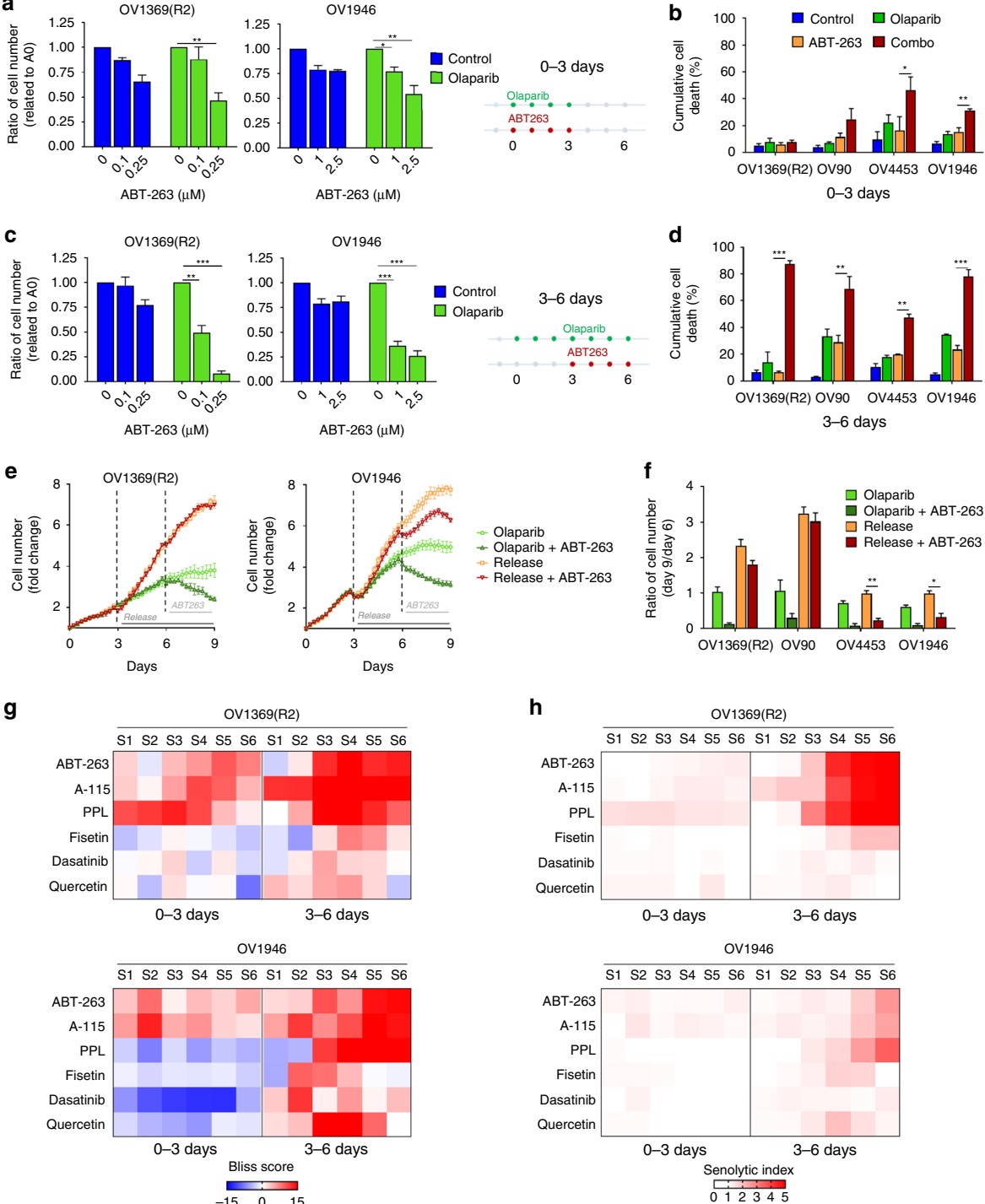

**Fig. 5** Bcl2/Bcl-XL inhibitors specifically target Olaparib-induced senescent cells. **a–d** Fold change in cell number (**a, c**) and percentage of dead cells (**b, d**) on day-6 of treatment regimens indicated in **a**, **c** (right panels). **e, f** Representative proliferation curves (**e**) or fold change in cell number (**f**) of HGSOC H2B-GFP cells treated with Olaparib for 9 days (light green), or for 6 days and sequential addition of ABT-263 for the next 3 days (dark green), or treated with Olaparib for 3 days and then without for the next 6 days (orange), or treated with Olaparib for 3 days and sequential addition of ABT-263 for the next 6 days (red). **g, h** Heat map of Bliss scores (**g**) or Senolytic Indexes (**h**) for combination treatments between Olaparib $IC_{50}$ concentrations and different concentrations of senolytics (see Supplementary Fig. 6D) for 3 days (0–3 days) or treated with Olaparib for 6 days with sequential addition of inhibitors after 3 days of Olaparib (3–6 days) of HGSOC cells expressing H2B-GFP. Data in (**e**) are representative curves of at least three independent experiments. For all the data, the mean ± SEM of three independent experiments is shown. Data were analyzed using the two-tail Student $t$ test. * Denotes $p < 0.05$, **$p < 0.01$, and ***$p < 0.001$

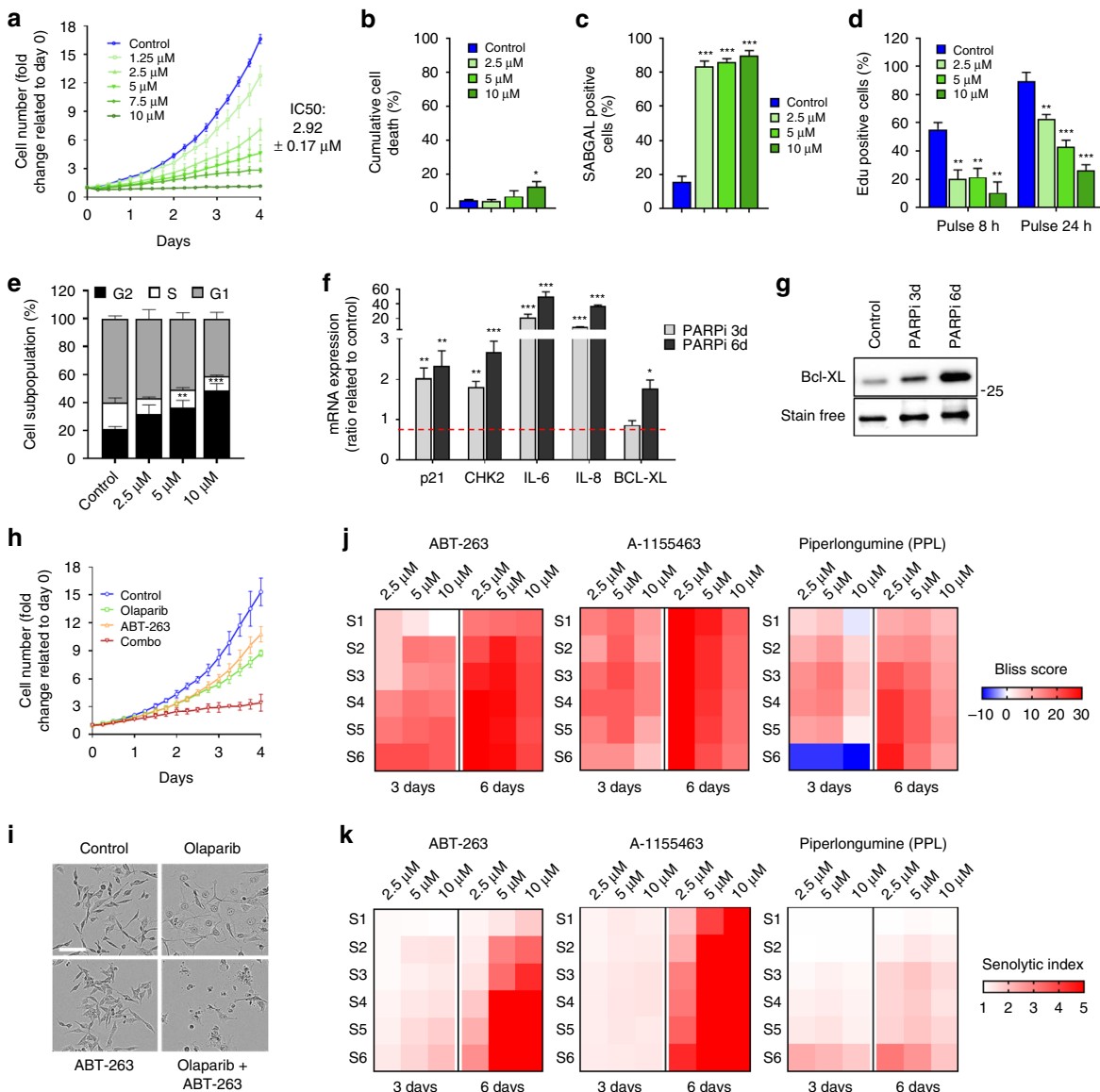

**Fig. 6** Olaparib induces a targetable senescence-like phenotype in a TNBC cell line. **a** Proliferation response of MDA-MB-231 cells treated with different concentrations of Olaparib for 6 days with analysis of cell numbers every 6 h. Control = nontreated. **b–e** MDA-MB-231 cells were treated with 2.5, 5, and 10 μM Olaparib for 6 days. **b** Cumulative cell death was analyzed by flow cytometry. **c** SAβgal positive HGSOC cells were evaluated. **d** Analysis of 8 and 24-h EdU pulse. **e** Flow cytometry analysis of cell cycle populations. **f** Relative mRNA levels of p21, CHK2, IL-6, IL-8, or BCL-XL evaluated by real-time Q-PCR in MDA-MB-231 cells treated with 5 μM Olaparib for 3 and 6 days. The values represent the fold change expression compared to nontreated controls. **g** Western blot detection of Bcl-XL in MDA-MB-231 treated with 5 μM Olaparib for 3 or 6 days. Total protein stain was used as a loading control. **h, i** Cell proliferation curves (**h**) and representative images (**i**) of MDA-MB-231 H2B-GFP cells treated with Olaparib (2.5 μM), ABT-263 (0.25 μM), or Olaparib/ABT-263 (2.5/0.25 μM). Scale bar, 150 μm. **j, k** Heat map of Bliss scores (**j**) or Senolytic Indexes (**k**) for combination treatments between Olaparib at 2.5, 5, or 10 μM concentrations and different concentrations of senolytics (see Supplementary Fig. 12A) for 3 days (0–3 days), or 6 days in TNBC cells expressing H2B-GFP. Data in (**a, g, h, i**) are a representation of at least three independent experiments. For all the data, the mean ± SEM of three independent experiments is shown. Data were analyzed using the two-tail Student $t$ test. * Denotes $p < 0.05$, **$p < 0.01$, and ***$p < 0.001$

_TP53_ mutant triple negative breast cancer (TNBC) MDA-MB-231 cells[41] revealed a concentration-dependent inhibition of cell proliferation that was in a $IC_{50}$-intermediate range when compared to HGSOC cells (Fig. 6a, $IC_{50}$: $2.92 \pm 0.17$ μM). As in HGSOC cells, Olaparib induced a senescence-like phenotype in MDA-MB-231 cells, including a very low cumulative cell death rate even at concentrations above the $IC_{50}$ (Fig. 6b, Supplementary Fig. 11A), a significant increase in SAβgal positive cells (Fig. 6c, Supplementary Fig. 11B), and a clear cell enlargement even at a lower concentration (2.5 μM) (Supplementary Fig. 11C, D). Short and long EdU pulse-labeling assays revealed a dose

dependent decrease in DNA synthesis at day 6 in Olaparib-treated TNBC cells (Fig. 6d), indicating an apparent and stable SAPA in MDA-MB-231 cells. This was confirmed by cell cycle analysis at 6 days post-treatment showing an accumulation at the G2/M phase of the cell cycle (Fig. 6e, Supplementary Fig. 11E). Furthermore, gene-expression analysis demonstrated that p21, CHK2, IL-6, IL-8, and BCL-XL were significantly upregulated in TNBC cells treated with Olaparib for 3 and 6 days (Fig. 6f, g). Thus, PARPi induced a significant senescent-like state with cell cycle arrest in TNBC cells. Importantly, a combination therapy of Olaparib at $IC_{50}$ or higher doses with the senolytics ABT-263,

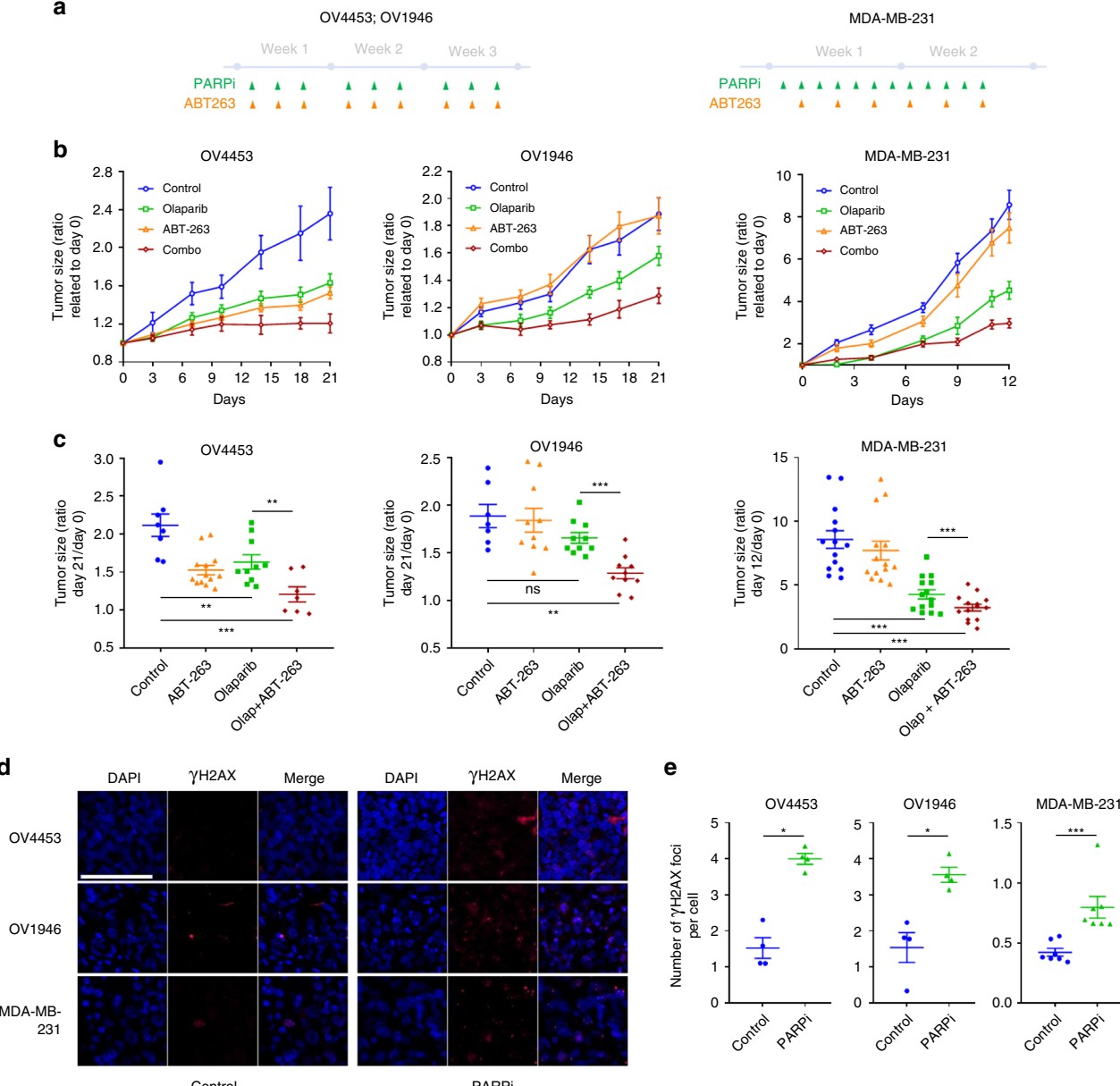

**Fig. 7** In vivo synthetic lethal interactions between PARP inhibition and cancer cell senescence. **a** Treatments time line of OV4453, OV1946 and MDA-MB-231. **b** Changes of OV4453, OV1946, and MDA-MB-231 tumor size after Olaparib (green, $n = 12$ for OV4453 and OV1946; $n = 16$ for MDA-MB-231), ABT-263 (orange, $n = 7$ for OV4453 and OV1946; $n = 16$ for MDA-MB-231), combination Olaparib + ABT-263 (red, $n = 10$ for OV4453 and OV1946; $n = 16$ for MDA-MB-231) or DMSO (blue, $n = 8$ for OV4453 and OV1946; n = 16 for MDA-MB-231). **c** Tumor size ratio between the first and the last day of treatment for each xenograft. **d, e** Representative images (**d**) and quantification (**e**) of γ-H2AX foci per nucleus in OV4453, OV1946, and MDA-MB-231 xenografts harvested two weeks after the first day of treatment. Scale bar, 100 μm. Data in (**c**) were analyzed using the two way ANOVA test with Bonferroni post-test correction. Complete statistics are supplied in Supplementary Tables 1–3. Data in (**e**) were analyzed using the two-tail Student $t$ test. *Denotes $p < 0.05$, **$p < 0.01$, and ***$p < 0.001$.

A-1155463, and to a lesser extent PPL had synergistic killing effects (Fig. 6h–k, Supplementary Fig. 12A–D), suggesting that the senescence-like state induced by PARPi therapy is common to ovarian and breast cancer cells and can be similarly targeted.

**In vivo synthetic lethal interactions between PARPi and senescent cancer cells.** In line with cell culture models, we found increased combination-treatment efficacy in pre-clinical models using xenografted HGSOC (OV4453, OV1946) and TNBC (MDA-MB-231) tumors (Fig. 7a–c). As PARPis are known to cause increased DNA damage-associated γH2AX foci in responding cancer cells[7], we validated the presence of γH2AX DNA damage foci in OV4453, OV1946 and MDA-MB-231 xenograft tumors treated with Olaparib (Fig. 7d, e). Furthermore, increased p21, CHK2, and Bcl-XL mRNA expression, but not that of Bcl2, were observed in MDA-MB-231 tumors harvested 12 days after initiation of Olaparib treatment (Supplementary Fig. 13). Taken together our in vitro and in vivo results allow us to propose a two-step model of targetable reversible senescence induced by PARPi as depicted in Fig. 8.

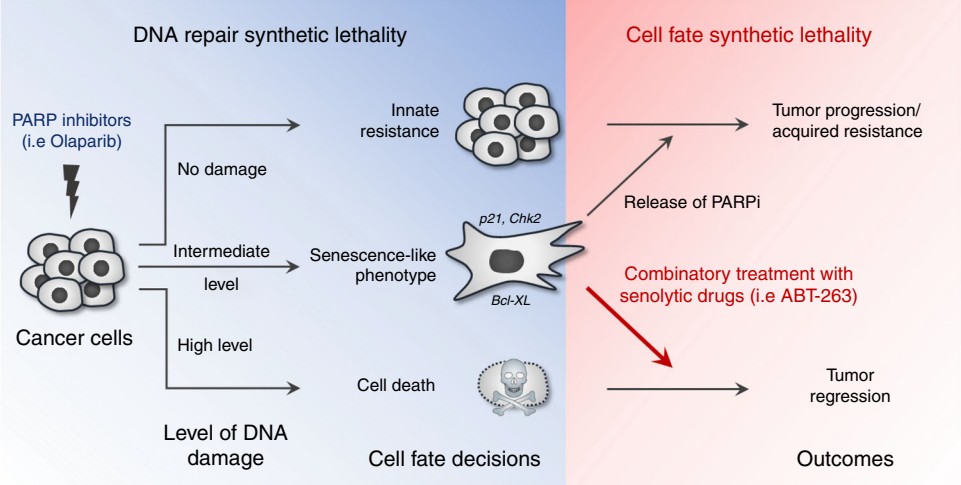

**Fig. 8** Model of targetable PARPi reversible senescence. The combination of PARPi and a senolytic was effective in preclinical models of ovarian cancer suggesting that coupling these synthetic lethalities provides a rational approach to their clinical use and may together be more effective in limiting resistance

## Discussion

Using HGSOC culture models, we find that PARPi triggers an accumulation of DSBs and persistent DDR signaling, which is accompanied by a reversible senescent-like G1 and G2/M cell cycle arrest. The CDKi activity of the p53 target p21 explains G1 accumulation of senescent-like HGSOC cells[42]. However, it is well-known that HGSOC display almost complete penetrance of *TP53* alterations (>95%)[22], as reflected for the *TP53* status of all the HGSOC cells used in this study[23–26]. It is possible that HGSOC cells behave like embryonic and oncogene-induced senescent cells where p53-independent increased levels of p21 have been reported[33]. Interestingly, FOXO3A has been shown to promote p53-independent p21/p27 transcription[43], and PARP inhibition is known to increases FOXO3A expression and activation, suggesting a potential mechanism for p21 activation during HGSOC-TIS[44]. Our own results support the involvement of the DDR kinase Chk2 in this p53-independent p21 upregulation induced by Olaparib. Chk2 activation has been shown to induce p21-dependent senescence in other p53 defective cells[34]. Chk2 also plays a central role in the DDR activated by DSBs where it is known to establish a G2 cell cycle arrest via its action on the CDC25-family of phosphatases[45], which is consistent with the PARPi-induced G2/M cell cycle arrest we have observed. In addition to chk2-p21 functions preventing early HGSOC cell death via mitotic catastrophe, we find that Bcl2-family members are gradually upregulated as HGSOC enter TIS, allowing PARPi-treated HGSOC cells to survive and develop a reversible senescent-like state.

Olaparib is recommended as maintenance therapy for ovarian cancer patients with germline *BRCA* mutations who responded favorably to chemotherapy treatments[46] and for prostate cancer patients with *BRCA* or *ATM* mutations[47]. Under this regimen, tumor cells are in constant exposure to PARPi. Based on our results, this approach may trigger a reversible cancer cell senescence-like state in patients, which directly explains tumor regrowth upon drug withdrawal, but may also underlie eventual treatment resistance. Resistance to Olaparib has been reported[48,49], and proposed mechanisms of resistance include the restoration of BRCA and/or HR functions by secondary mutations or copy number alterations in HR genes[48,49], or the increase in drug efflux regulation[5]. PARPi-induced HGSC-TIS provides an additional mechanism, whereby the constant release of SASP factors might provoke a niche of chemoresistant cells with altered

properties including increased stemness, which could support subsequent relapse. In line with this idea SASP can stimulate the proliferation of cancer cells in multiple ways including growth-related oncogenes in breast cancer and melanomas, and amphiregulin in prostate cancer[10,15,28]. Furthermore, matrix metalloproteinases composing the SASP promote migration of cancer cells while SASP IL-6 and IL-8, in addition to promoting tumor growth, have been reported to induce an epithelial to mesenchymal transition, thereby stimulating invasion and cancer stemness reprogramming[10,17,50]. In summary, our observation that HGSOC cells in culture re-grow almost immediately after PARPi removal suggests that TIS induced by these drugs in vivo can have an immediate reversible antitumoral effect via SA proliferation arrest but could have a long-term protumoral effect via the SASP.

Importantly, we demonstrated that PARPi-induced HGSOC-TIS is a "senescence-like" state that can be reverted in the majority of the cells within an otherwise senescent cell population, and not only in rare cellular clones that escape senescence as previously reported[9,11–14]. Understanding how and when cancer cells harbor this senescence-like state is important, as it provides a window of opportunity for additional pharmacological manipulation of TIS to benefit patients via enhanced treatment responses. For example, to avoid PARPi relapse and the emergence of chemoresistant cancer cells, we propose the removal of PARPi-induced senescent cells using senolytic drugs like ABT-263 (Navitoclax), an inhibitor of Bcl2 and Bcl-XL[18,20,21]. This drug has been extensively used in pre-clinical models and is the subject of past and ongoing clinical trials due to its pro-apoptotic effects[36]. In these studies and trials, ABT-263 was administered alone or in combination with chemotherapeutic agents (carboplatin/paclitaxel) or targeted therapies (such as Erlotinib) with the goal to directly sensitize damaged cells to apoptosis and cell death[36]. Our results indeed reveal a synergistic effect of PARPis and ABT-263 in all HGSOC and breast cancer cell lines tested, but importantly ABT-263 did not induce immediate apoptosis and cell death when administered in simple combination with, or after removal of PARPi, revealing that instead of a direct synergy, combo-efficacy with PARPi is achieved via specific targeting of the senescent-like state produced under prolonged PARPi exposure. We thus propose a two-step synergistic working model in which PARPi must first create DNA damage and senescence, a cellular state that can be subsequently targeted for senolysis

(Fig. 8). Although we present direct evidence for a two-step approach to exploit interconnected DNA repair-senescence synthetic lethalities, the model predicts that PARPi therapies (first drug) can be enhanced at two levels, first via additional DNA damage and cell cycle-related synthetic lethal approaches (second drug), which could create more DNA damage and senescent cells that are then targetable via senolysis (third drug). A recent study focusing on the pro-apoptotic role of Navitoclax reported a similar synergistic effect with Talazoparib in ovarian cancer cells[51] and our results using a panel of senolytic agents now support the notion that this was caused by a senolytic effect. Importantly, here we demonstrate that this therapy-induced p53-independent senescence-like state is not restricted to HGSOC since similar results are observed using *TP53* mutant TNBC cells (MDA-MB-231) including in vivo, and we show that Bcl-XL is the most relevant Bcl2 family member in this context. Cellular senescence can be a potential mechanism of chemoresistance in breast cancer[52], suggesting that senescence-targeting combination-therapies may represent promising alternatives for both HGSOC and TNBC.

Overall, the present work shows that PARPis induce a reversible p53-independent senescence-like state that can be combined to senescence-targeting drugs to potentiate treatment effects (Fig. 8). Unlike the majority of combination therapies using PARPis, this strategy does not involve additional DNA damage-based synthetic lethality, but rather targets the cell fate decisions induced by these inhibitors.

## Methods

**Cell lines and cell culture**. The four human HGSOC cell lines used, OV1369(R2), OV90, OV4453, and OV1946, were derived in our laboratory from the ascites of patients diagnosed with HGSOC and have been extensively characterized[23–26]. All cell lines were maintained in a low oxygen condition (7% $O_2$ and 5% $CO_2$) and grown in OSE medium (Wisent, Montreal, QC) with 10% fetal bovine serum (FBS) (Wisent), 0.5 μg ml$^{-1}$ amphotericin B (Wisent) and 50 μg ml$^{-1}$ gentamicin (Life Technologies Inc., Burlington, ON). The MDA-MB-231 breast cancer cell line was a gift from the laboratory of Dr. John Stagg (CRCHUM, Canada) which was purchased from ATCC. It was maintained in DMEM (Wisent) with 10% FBS (Wisent), 0.5 μg ml$^{-1}$ amphotericin B (Wisent) and 50 μg ml$^{-1}$ gentamicin (Life Technologies Inc.). All HGSOC cell lines were authenticated in 2017 using short tandem repeat (STR) profiling by the McGill University Genome Center (Montreal, Canada). All cell lines were tested negative for mycoplasma with IDEXX BioAnalytics (Columbia, MO65201).

**IncuCyte™ cell proliferation phase-contrast imaging assay**. For cell proliferation evaluation, 1500 cells per well were seeded for OV1369(R2), OV90, OV1946, and MDA-MB 231, and 2500 cells per well were seeded for OV4453 in 96-well plates (all H2B-GFP expressing). Cells were incubated with PARPi and senolytic drugs at different concentrations and times in adjusted 0.75% dimethyl sulfoxide (DMSO). Cell number was imaged by phase contrast and fluorescence using the IncuCyte™ Live-Cell Imaging System (IncuCyte HD). Frames were captured at 6-h intervals from two separate regions per well using a 10× objective. Proliferation growth curves were constructed using IncuCyte™ Zoom software from H2B-GFP cell nuclei number measurements. IC$_{50}$ values were determined by using GraphPad Prism 6 software (GraphPad Software Inc., San Diego, CA). Each experiment was performed in triplicate and repeated three times.

**Clonogenic assays**. Cells were seeded in a 6-well dish at a density of 1000 cells per well and allowed to adhere for 16 h in a 37 °C, 5% $CO_2$, 7% $O_2$ incubator after which the media was removed and replaced with OSE complete media containing Olaparib. Cells were treated for 6 days or 12 days or treated for 6 days then incubated in a drug-free medium for another 6 days and further incubated until colonies became visible at a ×2 magnification. Cells were fixed with cold methanol and colored with a mix of 50% v/v methanol and 0.5% m/v blue methylene (Sigma-Aldrich Inc., St. Louis, MO). Colonies were counted under a stereomicroscope and reported as a percentage of control. Each experiment was performed in triplicate and repeated three times.

**Immunofluorescence**. Cells were seeded onto coverslips in 12-well plates and grown for 3 and 6 days. Cells were fixed in formalin for 10 min at room temperature (RT) and permeabilized in 0.25% Triton in phosphate-buffered saline (PBS) for 10 min. Slides were blocked for 1 h in PBS containing 1% bovine serum albumin (BSA) and 4% donkey serum. Primary antibodies diluted (1/2500 for

γH2AX and 53BP1) in blocking buffer were added and slides were incubated overnight at 4 °C. Cells were washed and incubated with secondary antibodies (dilution 1/5000) for 1 h at RT, then washed again. Coverslips were mounted onto slides using Prolong® Gold anti-fade reagent with DAPI (Life Technologies Inc.). Images (×400 magnification) were obtained using a Zeiss microscope (Zeiss AxioObserver Z1, Carl Zeiss, Jena, Germany). Automated analysis software from Zeiss (AxioVision™, Carl Zeiss) was used to count foci to calculate the average number of foci per nucleus. The fold change was calculated as the ratio between percentages of γH2AX or 53BP1 nuclear foci in treated versus control (nontreated) cells. γH2AX and 53BP1 foci were quantified in >150 nuclei from 3 different fields of each coverslip.

**EdU (5-ethynyl-2′-deoxyuridine) detection**. To detect DNA synthesis, cells were seeded onto coverslips in 12-well plates. Drugs were added 24 h after seeding. EdU (10 μM, Invitrogen) was added to the medium and incubated for 8 or 24 h before the end of drug treatment on days 3 or 6. Cells were washed three times with TBS and fixed with 10% formalin for 10 min. EdU staining was assessed using the Click-iT® EdU Alexa Fluor® 488 Imaging Kit (Invitrogen). Coverslips were mounted onto slides using Prolong® Gold anti-fade reagent with DAPI (Life Technologies Inc.). Images were obtained using a Zeiss microscope (Zeiss AxioObserver Z1, Carl Zeiss, Jena, Germany). Automated analysis software from Zeiss (AxioVision™, Carl Zeiss) was used to count foci.

**Analysis of cell cycle and cell death by flow cytometry**. Cells were seeded in 6-well plates and treated 24 h after seeding, then harvested 3 and 6 days after. For cell cycle analysis, live cells were fixed for 24 h in 70% ethanol and incubated for 30 min at RT with 100 μg ml$^{-1}$ RNAse A and 25 μg ml$^{-1}$ propidium iodide. For cell death analysis, all the cells were incubated 30 min at RT with BV421 AnnexinV (563973, BD Biosciences, San Jose, CA) (dilution 1/10) and 5 min at RT with 0.9 nM of DRAQ7 (ab109202, Abcam Inc.). A maximum of 10,000 events were counted per condition using the Fortessa flow cytometer (BD Biosciences, Mississauga, ON) and analyzed with FlowJo software.

**SA β-galactosidase detection**. We adapted the SA β-galactosidase assay originally described by Dimri et al.[29] Briefly, cells grown in 6-well plates were washed once with ×1 PBS and fixed with 10% formalin for 5 min, then washed again with PBS and finally incubated at 37 °C for 6–24 h (depending on the cell line) in a staining solution composed of 1 mg ml$^{-1}$ 5-bromo-4-chloro-3-inolyl-β-galactosidase in dimethylformamide (20 mg ml$^{-1}$ stock), 5 mM potassium ferricyanide, 150 mM NaCl, 40 mM citric acid/sodium phosphate, and 2 mM MgCl$_2$, at pH 6.0. Afterward, cells were washed twice with PBS and pictures were taken for quantification.

**Drug combination analysis**. HGSOC cells were infected with H2B-GFP. Synergistic combinations were evaluated using the Chou-Talalay method[53], using CompuSyn software (ComboSyn Inc., Paramus, NJ, USA) with constant ratios of drug combinations [75/1 for OV1369(R2), 8/1 for OV90, 0.1/1 for OV4453, and 0.1/1 for OV1946]. The resulting CI values defined synergistic (<0.9), additive (0.9–1.1), and antagonistic (>1.1) effects in drug combinations. The Bliss independence model[38] was also used to assess combination activity, with negative integers indicating antagonism, a value of zero indicating additive activity, and positive integers indicating synergy. Bliss scores were calculated for each combination in the dose matrix (1 × 6, one dose of Olaparib and 6 doses of one of the following: ABT-263, ABT-199, A-133, A-115, Piperlongumine, Fisetin, Dasatinib, or Quercetin) and totaled to give a "Bliss sum" value. In addition, for each dose of senolytic, the SI was calculated as follow: SI = [(Nf/Ni) SX/(Nf/Ni) S0] No PARPi/ [(Nf/Ni) SX/(Nf/Ni) S0] PARPi, where Nf = final cell number; Ni = initial cell number; SX = X dose of the senolytic; S0 = no senolytic.

**Murine xenograft model**. All animal experiments complied with all relevant ethical regulations for animal testing and research at CRCHUM. All experiments were done with approval from our institutional committee on animal care (CIPA) under the protocol number C14008AMMs. Cells in exponential phase were prepared at a concentration of 7.5 million cells/ml for OV1946 or OV4453 and 5 million cells/ml for MDA-MB-231 in 50% PBS −50% matrigel. NRG mice (NOD-Rag1$^{null}$ IL2rg$^{null}$, NOD rag gamma) were obtained from the Jackson laboratory (Bar Harbor, ME). All experiments were carried out with 6-week-old females. To initiate tumor xenografts, 0.5 mL of cell suspension was injected into the right and left flanks. Mice were weighed and tumor volumes measured twice per week. Forty days post-injection mice with similar tumor size (400–500 mm$^3$) were randomized based on their weight into four groups: untreated, treated with Olaparib, treated with ABT-263, or treated with both. Olaparib (50 mg kg$^{-1}$) was administered intraperitoneally (100 μl per injection) and ABT-263 (50 mg kg$^{-1}$) was administered orally by gavage (100 μl per gavage) every Monday, Wednesday and Friday for OV4453 and OV1946. For MDA-MB-231 Olaparib was injected every day and ABT-263 every 2 days. Olaparib and ABT-263 were similarly administered to control groups. Olaparib vehicle was PBS with DMSO 12%. For mouse models ABT-263 stock was prepared at 100 mg ml$^{-1}$ and vehicle was ethanol:polyethylene glycol 400 (Sigma Aldrich, St. Louis, MO): Phosal 50 PG (Lipoid (Germany) cat#368315) at a ratio of 1:3:6. Mice were sacrificed when tumor volumes increased

two times from the day of injection (800–1000 mm$^3$) or after two weeks of treatments. Animals that had health issues before these endpoints were excluded from the analyzes. Tumor growth plots and survival curves were built using the GraphPad Prism 6 software (GraphPad Software Inc.).

**Drugs.** Olaparib (AZD2281), ABT1155463 (S7800), ABT-1331852 (S7801), and Dasatinib (S1021) were purchased from Selleckchem (Houston, TX), ABT-263 (Navitoclax) and ABT-199 (A8194) from APExBIO (Houston, TX), Niraparib (M2215) from AbMole Bioscience (Houston, TX), Talazoparib (HY-16106) from MedChem Express (Monmouth Junction, NJ), Piperlongumine (1919) from Bio-Vision (Milpitas, CA), Fisetin (15246) from Cayman chemical (Ann Arbor, MI), and Quercetin (Q-4951) from Sigma Aldrich (St. Louis, MO). Drugs were dissolved in 100% DMSO and then further diluted in complete culture media for in vitro experiments. Drugs were added 24 h after seeding.

**Antibodies.** The following antibodies were used: Bcl-XL (clone 54H6) (2764 s; Cell Signaling, Danvers, MA) (dilution for western blot 1/2000); Bcl-2 (C124) (M0887; Dako, Agilent) (dilution for western blot 1/1000); phospho-histone γ-H2AX (clone JBW301, EMD Millipore, Temecula, CA) (dilution for immunofluorescence 1/2500); 53BP1 (clone 305, Novus Biologicals, Littleton) (dilution for immunofluorescence 1/2500).

**Cloning, viruses, and infections.** Viruses were produced as described previously and titers were adjusted to achieve ~90% infectivity[54]. Infections were followed 48 h later by puromycin or hygromycin selection and stable cells were either used or stored at −80 °C. Lentiviruses encoding H2B-GFP were produced by amplification of the H2B sequence from the pENTR1A-H2B-HcRed plasmid (a gift from the laboratory of Dr. Richard Bertrand, CRCHUM, Canada) using the following primers: ES-92 and ES-93. The amplified product was inserted into the entry vector pENTR1A-GFP-N2 (FR1) following digestion with the restriction enzymes Kpn1/BamH1 (pENTR1A-H2B-GFP), which was then recombined into the lentiviral destination vector (pLenti PGK Hygro Dest (W530-1)) to obtain the final lenti-H2B-GFPhygro lentivector[55]. The shRNAs lentivectors against p21, Chk2, and RFP (red fluorescent protein, control) were purchased from Open Biosystems (lentiviral PLKO.1 vector with puromycin selection).

**Protein preparation and western blot analysis.** Whole-cell lysates were prepared by scraping cells with mammalian protein extraction reagent (MPER, Thermo Fisher Scientific, Waltham, MA) containing a protease and phosphatase inhibitor cocktail (Sigma-Aldrich Inc., St. Louis, MO). Protein concentration was measured using the bicinchoninic acid (BCA) protein assay (Thermo Fisher Scientific). Ten micrograms of total protein extract were separated in stain-free 4–15% gradient Tris-glycine SDS-polyacrylamide gels (Mini PROTEAN® TGX Stain-Free™ Gels, Bio-Rad Laboratories, Hercules, CA) and transferred onto PVDF membranes (Hybond-C Extra, GE Healthcare Life Sciences, Mississauga, ON, Canada). Membranes were blocked with 5% BSA in PBS for 1 h and probed with primary antibodies overnight at 4 °C. Bound primary antibodies were detected with peroxidase-conjugated secondary antibodies (Santa Cruz Biotechnology Inc., Dallas, TX) and enhanced chemiluminescence (Thermo Fisher Scientific). Chemiluminescence was detected using the ChemiDoc MP Imaging System Bio-Rad Laboratories). The control of the loading protein was evaluated using the stain-free technology (Bio-Rad Laboratories). The original uncropped and unprocessed images of western blot and stain free membranes are supplied in Supplementary Fig. 14.

**Real-time Q-PCR.** Total RNA was extracted from harvested cells or from MDA-MB-231 flash-frozen tumors using the RNeasy kit (Qiagen Inc., Hilden, Germany). One microgram of total RNA was subjected to reverse transcription using the QuantiTect Reverse Transcription Kit (Qiagen Inc.). One microliter of the reverse-transcribed product was diluted (1:10) and subjected to Q-PCR using sequence specific primers (400 nM) and the SYBR Select Master Mix (Applied Biosystems®, Life Technologies Inc.). Sequence primers for target genes p21, p27, p15, p16, p57, IL8, Bcl2, Bcl-XL, and CHK2 are described in the "List of primers" section. Q-PCR were performed using Applied BioSystems® Step One Plus apparatus (UDG activation 50 °C/2 min, followed by AmpliTaq activation plus denaturation cycle 95 °C/2 min, followed by 40 cycles at 95 °C/15 s, 60 °C/1 minute and 72 °C/30 s). Gene expression values were normalized to TATA-binding protein gene expression. Three independent experiments were performed in duplicate.

**Cytokine secretion measurement in conditioned medium.** Conditioned media (CM) were prepared by incubating cells with OSE medium without FBS for 24 h and stored at −80 °C until probed. Levels of IL-6 and IL-8 were assessed using ELISA (R&D Systems (IL-6 #DY206; IL-8 #DY208)). The data were normalized to cell number and reported as fold change of secreted protein compared to the control. CM of OV1369(R2), OV90 and OV1946 were also assayed using multiplex-40 ELISA MSD V-PLEX Products: Proinflammatory Panel I, Chemokine Panel I, Angiogenesis Panel I, Cytokine Panel I and Vascular Injury Panel II (MSD,

Gaithersburg, MD). The data were normalized to cell number and reported as log 2 —fold change of secreted protein compared to the control.

**Immunohistochemistry on xenograft tissue.** Tumors were harvested 2 weeks after the beginning of treatments and were formalin-fixed and paraffin-embedded. Four micrometer thick paraffin-embedded tissue sections were cut and immunostained using an automated Ventana Discovery XT staining system (Ventana Medical Systems). Antigen retrieval was performed in cell conditioner 1 and slides were incubated with anti-phospho-histone γ-H2AX antibody (1:250 dilutions in PBS at 37 °C for 60 min, clone JBW301, EMD Millipore, Temecula, CA) on automatic. Then, on the bench, slides were incubated for 20 min with blocking solution (Dako, Agilent, #X0909) followed by incubation with the secondary antibody (1:250 dilution in PBS at RT for 45 min; anti-mouse Cy5, Life Technologies Inc., #A10524). Finally, slides were incubated 15 min at RT with a 0.1% (w/v) solution of Sudan Black in 70% ethanol to quench tissue auto fluorescence. Slides were mounted using ProLong® Gold Antifade Mountant with DAPI (Life Technologies Inc., P-36931). Between each step, except after the blocking steps, slides were washed twice with PBS. Slides were stored at 4 °C and scanned the next day. A negative control slide was done in parallel where PBS replaced the primary antibody.

**TCGA dataset.** Gene microarray expression (Affymetrix U133) for BCL2 and BCL2L1 (Bcl-XL gene) were directly downloaded from the cBioportal for Cancer Genomics website (www.cbioportal.org)[56,57] using the high-grade serous ovarian cancer cohort (n = 530) from The Cancer Genome Atlas (TCGA)[22]. Graph Pad Prism 6 was used to plot the data and to perform statistical analyzes using the unpaired Mann–Whitney test.

**List of primers.** ES-92: 5′-GGTACCCCACCATGCCAGAGCCAGCGAAGTC TGCT-3′
ES-93: 5′-GGATCCTAGCGCTGGTGTACTTGGTGATGG-3′
shp21.4 (#TRCN0000040124): 5′-GCTGATCTTCTCCAAGAGGAA-3′
shp21.6 (#TRCN0000040123): 5′-CGCTCTACATCTTCTGCCTTA-3′
shCHK2-12 (#TRCN0000010314): 5′-ACTCCGTGGTTTGAACACGAA-3′
shChk2.1 (#TRCN0000039946): 5′-GCCAATCTTGAATGTGTGAAT-3′
shRFP (#TRCN0000231725): 5′-ACTACACCATCGTGGAACAGT-3′
p21: 5′-GGGACAGCAGAGGAAGAC-3′ F; 5′-TGGAGTGGTAGAAATCT GTCA-3′ R
p27: 5′-TCCGGCTAACTCTGAGGACA-3′ F; 5′-GTAGAAGAATCGTCGGT TGC-3′ R
p15: 5′-GAATGCGCGAGGAGAACAAG-3′ F; 5′- CATCATCATGACCTGGA TCGC-3′ R
p16: 5′-GGGAGCAGCATGGAGCCT-3′ F; 5′-ATGACCTGGATCGGCCT CCGACCGT-3′ R p57: 5′-GCGGTGAGCCAATTTAGAGC-3′ F; 5′-CGGTTGCT GCTACATGAACG-3′ R
IL8: 5′-GCCAACACAGAAATTATTGTAAAG-3′ F; 5′-TTATGAATTCTC AGCCCTCTTC-3′ R
Bcl2: 5′-AACATCGCCCTGTGGATGAC-3′ F; 5′GGCCGTACAGTTCCA CAAAG-3′ R
Bcl-XL: 5′GGCCACTTACCTGAATGACC-3′ F; 5′AAGAGTGAGCCCAGCA GAAC-3′ R
CHK2: 5′-AAGCTAAATCATCCTTGCATC-3′ F; 5′AGCTTCTTTCAGGCG TTTATTC-3′ R

**Statistics.** Unless noted otherwise, statistical analyses were performed using the two-tail Student t test, which was appropriate for most experimental designs as the data was normally distributed and the variance between groups that were being statistically compared was similar. All in vitro experiments were repeated at least three times (n = 3) and mouse tumor numbers are indicated for every cohort.

**Reporting summary.** Further information on research design is available in the Nature Research Reporting Summary linked to this article.

## Data availability
Source data for the statistics presented in the figures is provided as a Source Data file. Additional relevant source data supporting this study is available from the corresponding authors upon request.

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

## Acknowledgements

We thank Mes-Masson and Rodier laboratory members for valuable comments and discussions, and Jacqueline Chung for manuscript editing. We thank the CRCHUM animal facility and the Institut du cancer de Montréal (ICM) Imaging and Live imaging platform. We thank Pamela Thebault from the ICM FACS platform. We thank the Molecular Pathology core facility of the CRCHUM for performing the

immunohistochemistry on xenograft tissue. This work was supported by the ICM (D.P., A.M.M.M., and F.R.) and by the Canadian Institute for Health Research (CIHR MOP114962 to F.R.), the Terry Fox Research Institute (TFRI 1030 to F.R.) and the Cancer Research Society (CRS) in partnership with Ovarian Cancer Canada (20087 to A.M.M.M., D.P. and 22713 to F.R.). This work was also supported by the Oncopole in collaboration with the Fonds de recherche du Québec - Santé (FRQS), CRS, Génome Québec and IRICoR (265877 to D.P., A.M.M.M and F.R.). A.M.M.M., D.P. and F.R. are researchers of CRCHUM/ICM, which receive support from the FRQS. F.R is supported by a FRQS junior I-II career awards (22624, 33070). Ovarian tumor banking was supported by the Banque de tissus et de données of the Réseau de recherche sur le cancer of the FRQS affiliated with the Canadian Tumor Repository Network (CTRNet). H.F. received the ICM Michèle St-Pierre Bursary. N.M. was supported by a MITACS fellowship. H.F., A.M. and S.A.S. received Canderel fellowships from the ICM.

## Author contributions

H.F., N.M., and F.R. conceived and designed the experiments. H.F., N.M., A.M., S.A.S., V.T., S.G., M.A.O., and L.C. conducted the experiments. K.L.D. conducted in vivo experiments. E.C., D.P., A.M.M.M., and F.R. supervised the project. H.F., N.M., E.C., A.M.M.M., and F.R. wrote the paper.

## Additional information

**Competing interests:** The authors declare no competing interests.

