## [Peer Review File · Nature Communications]

Reviewers' comments:

Reviewer #1 (Remarks to the Author):

1) What are the major claims of the paper?

The main claim of the manuscript is that PARPi (Olaparib) treatment results in cell death to the extent of 20 or 40% in PARPi-resistant or PARPi-sensitive high grade serous ovarian cancer cells, respectively. The remaining cells show loss of proliferation capacity and senescence-associated phenotype (larger size, SA-b-gal, IL6 or IL-8 expression, etc) that can be reversed if PARPi is withdrawn. Treating these PARPi-treated cells that have established senescence phenotype with Bcl2/BCL-XL-targeting senolytic drug (ABT-263) results in synergistic cell killing in PARPi-resistant and sensitive cells in vitro. The major claim based on these results is that PARPi plus senescence inhibitor works synergistically for PARPi sensitive or resistant tumor cells (in vitro).

However, the in vivo animal tumor model, the drug ABT-263 is effective with PARP1 only for PARPi-sensitive cell lines OV4453 and OV1946 but not for PARPi-resistant cell lines. This discrepancy between in vitro and in vivo results led to two new arguments: (a) specific inhibitors of BCLXL rather than BCL2-inhibitor would work better for killing PARPi-resistant senescent tumor cells; or (b) other senolytic drugs that target diverse senescence pathways could kill both PARPi-resistant OV1369 and PARPi-sensitive OV1946 cell lines. These two concepts were shown to work in in vitro models but no data was shown for these two ideas to work in vivo animal tumor model.

Finally, other PARPi also induce senescence in ovarian cancer cells and reversible senescence was noted in breast cancer cells too. In short, the authors propose that results are universally applicable.

2) Are the claims and findings novel and will they be of interest to others in the community and wider field?

A previous paper Yokoyama et al. (2017) (ref 51 of the manuscript) has shown that PARPi (BMN673 or niraparib) with ABT263 synergistically cause more apoptosis in high grade ovarian cancer cell lines, such as OVCAR3, 8 and OV90. Since OV90 cell line is common in two studies and since end-result is more cell death, there is an overlap of results between two studies. However, since Yokoyama group did not attempt to characterize the PARPi-treated cells as senescent, the current manuscript could be accepted to have made a novel mechanistic explanation that senescence induced by PARPi treatment is the target for enhanced killing by senescence targeting drugs.

Moreover, Chang et al. 2016 (Nature Medicine, 22, 79-83) had already identified that ABT263 kills senescent cells via apoptosis. Thus, while Yokoyama et al failed to connect this effect of ABT263, this manuscript effectively exploited this interpretation to make a clear and novel mechanistic argument of targeting senescent cells. Therefore, this finding would be of interest to the community and wider field.

3) Is the work convincing?

This is the weakest link in the paper. There is too much of data (2 PARPi-resistant and 2 PARPi-sensitive cell lines, variety of assays, and plethora of data which is grossly generalized across the board for all cell lines despite significant differences in results.

For example, 20% to 40% apoptosis is classed as similar level of death, or IL2 and IL-8 differences among cell lines are glossed over. The differing degree of Knockdown for shp21 (50%) and shchk2 (80%) are treated at par KD level although extent of cell death varies between these two cell lines for each KD. The discordance between mRNA and protein levels for IL2 and 8 is glossed over. For many other parameters, only mRNA is analyzed without a mention of what are the actual differences in protein levels in these cells. The lack of proper explanation for data variability among different figures erode the value of global data.

4) Some big picture concepts relevant for manuscript have not been addressed:

A) If total death by combination of two drugs is due to PARPi-induced direct death and ABT263-induced death of senescent cells (Fig. 4C), it should be expected that a sum of % of dead and senescent cells by PARPi alone as shown in Figs 1B and 1D respectively, should be reflected in total death caused by PARPi plus ABT263 (Fig. 4C).

However, this is not the case even at the lowest dose of PARPi in Fig 1B or D, which is expected to have been applied in Fig 4C. For example, for OV1369, 10% death and 20% senescence at low PARPi in Fig. 1, becomes 70% total death in Fig. 4C. There is no accounting for 40% additional death. For Ov1946, 45% death and 10% senescence at lowest PARPi in Fig.1 becomes 70% total death in Fig. 4.

No efforts have been made to account for additional death (synergistic effect) of two drugs, which could be more in line with Yokoyama data-explanations as more death due to more pro-death environment created by ABT263.

B) The reason for using PARPi-resistant versus PARPi-sensitive cell lines is not emerging in any of the data, as no effort has been made to correlate resistance with eventual response. An intuitive assumption could be that PARPi-resistant cells would have more senescence (as an explanation of resistance) and therefore more profound effect of ABT263 co-admin with PARPi. But no such effect was anticipated, observed or explained.

C) The in vivo tumor data (Fig. 4F) shows combo PRRPi+ABT263 being effective only in PARPi-sensitive cell lines (4453 and 1946) and no data is shown for any of the two PARPi-resistance cell lines (1369 and 90). If we assume that for resistant cells, tumor data of combo treatment did not produce dramatic results, it would have been better to show animal data and then explain, as it was tried to be done in Fig 4G.

On the other hand, the authors just add more data in vitro with other Senolytic agents (acting on different targets) and continue to make the argument with more in vitro data about universality of the combination effectiveness for PARPi-resistant AND PARPi-sensitive cell lines, while completely ignoring that fact that this did not work in vivo for PARPi-resistant cell lines.

The data series for other senolytic agents added after Fig 4 appears weak without support from in vivo tumor data.

In summary, lack of strong support for the model in the in vitro data with mouse model is glossed over and not explained.

5) What further evidence would be required to strengthen the conclusions?

Manuscript needs extensive revision with streamlined and consistent arguments strongly supported by in vitro and supporting in vivo data to make a strong case of the main argument.

6) Would the paper influence thinking in the field?

Yes, if the conclusions are strongly supported by consistent and clear data.

7) Other questions and concerns?

Great potential for the therapeutic application of the work.

8) Appropriateness and validity of statistical analyses, if applicable.

Could have shown more of raw data to support graphs and charts shown as the main data.

Reviewer #2 (Remarks to the Author):

Manuscript#: NCOMMS-18-14775

Corresponding Author: Francis Rodier

Title: Exploiting interconnected synthetic lethal interactions between PARP inhibition and cancer cell reversible senescence

Summary: This is an extremely well-written manuscript that rigorously explored the relationship of PARP inhibitors and senescence in high-grade serous ovarian carcinoma (HGSOC) cell lines. The major findings of the work show for the first time a P53-independent, treatment-induced senescence-like phenotype in HGSOC cells following treatment with PARPi that is reversible upon drug withdrawal and that could be further exploited by the addition of senolytic drugs to achieve synergistic cell killing. The authors nicely show PARPi-induced acquisition of senescence-associated features (e.g., increased cell size, decreased proliferation, cell cycle arrest, SA- β -gal expression, IL-6 and IL-8 secretion and limited apoptosis accompanied by increased expression of Bcl-2 and Bcl-XL) in multiple PARPi-resistant and sensitive cell lines following olaparib treatment. They further show the recovery of cells following drug withdrawal, supporting reversal of the senescence phenotype in a significant component of the cell population, particularly the more PARPi-resistant cell types. They then showed the vulnerability of PARPi-treated cells to senolytic agents targeting Bcl-2 and/or Bcl-XL in vitro, demonstrating a prominent role of Bcl-XL in treatment-induced senescence and synergistic cell killing when olaparib was combined with ABT-263 or two additional compounds selective for Bcl-XL (A115 and A133). The enhanced effect of combining olaparib and ABT-263 were then confirmed in vivo in a xenograft model, and the synergistic effects of ABT-263 in vitro were similar with other PARPi (niraparib and talazoparib). Moreover, similar effects were shown in TP53 mutant triple negative breast cancer cells, suggesting potential efficacy in treatment of breast cancer. Overall, this is a well-designed and executed study on a topic of broad interest to the cancer research community, describing novel mechanism that may contribute to the emergence of PARPi resistance, and potential clinical impact ovarian and breast cancer patients.

Additional comments/suggestions are provided below that might be considered to improve rigor and/or clarity:

1. Experiments shown in Figure 2 (Figure 2D-H) testing for CHK2-p21 mediated regulation of proliferation arrest, employ a single shRNA for depletion of p21 and Chk2. The specificity of the observed effects on cell number, % cell death, DNA damage and mitotic catastrophe would be best supported/confirmed with independent shRNA for each gene and is commonly expected/required for publication in high quality journals.
2. Figure S4 is shown to establish the increase in expression of anti-apoptotic genes Bcl-XL and Bcl-2. The mRNA expression levels support an increase in expression of both Bcl-XL and Bcl-2, but

the WB shown for Bcl-XL protein expression (Fig. S4C) shows little difference in 3/4 cell lines - only OVC1946 shows clear/convincing increase in protein the parallels increase mRNA expression. The authors should comment on the effect of drugs in the absence of changes in protein expression.

3. In regard to the ABT-263 and Olaparib synergy, and possibly relevant to the point above, the concentration of ABT-263 used seems high (micromolar concentrations). What are EC50s for ABT263 in HGSOC cells, what confirmation if any is there that at micromolar concentrations of ABT-263 that effects remain on-target? The same question about off-target effects applies other senolytic agents used at high concentration.

4. Experiments showing the in vivo effects (Figure 4) show xenografts for cell lines demonstrated to be sensitive to Olaparib in vitro. It would be of clinical interest/importance to know if the combination treatment show in vivo activity in the more PARPi-resistant cells, as this may be a means to improve PARPi sensitivity to benefit patients who might otherwise not be expected to benefit from this class of drugs. Additional information could be provided from the analysis of the tumors collected (Fig. 4F), such as tumor morphology and evidence of senescence, apoptosis and on-target effects of olaparib and ABT-263.

5. Evaluation of combination therapy in an MDA-MB-231 xenograft model would further support the in vitro results and the potential clinical application in breast cancer.

November 15, 2018

Exploiting interconnected synthetic lethal interactions between PARP inhibition and cancer cell reversible senescence

Hubert Fleury*, Nicolas Malaquin*, Véronique Tu, Sophie Gilbert, Aurélie Martinez, Marc-Alexandre Olivier, Alexandre Sauriol, Laudine Communal, Kim Leclerc-Desaulniers, Euridice Carmona, Diane Provencher, Anne-Marie Mes-Masson# and Francis Rodier# (* These authors contributed equally; # corresponding authors)

Point by point response:

Reviewer text in blue

Responses in black

Reviewer #1 (Remarks to the Author):

1) What are the major claims of the paper?

The main claim of the manuscript is that PARPi (Olaparib) treatment results in cell death to the extent of 20 or 40% in PARPi resistant or PARPi sensitive high grade serous ovarian cancer cells, respectively. The remaining cells show loss of proliferation capacity and senescence associated phenotype (larger size, SAbgal, IL6 or IL8 expression, etc) that can be reversed if PARPi is withdrawn. Treating these PARPi treated cells that have established senescence phenotype with Bcl2/BCLXL targeting senolytic drug (ABT263) results in synergistic cell killing in PARPi resistant and sensitive cells in vitro. The major claim based on these results is that PARPi plus senescence inhibitor works synergistically for PARPi sensitive or resistant tumor cells (in vitro).

However, the in vivo animal tumor model, the drug ABT263 is effective with PARP1 only for PARPi sensitive cell lines OV4453 and OV1946 but not for PARPi resistant cell lines. This discrepancy between in vitro and in vivo results led to two new arguments: (a) specific inhibitors of BCLXL rather than BCL2inhibitor would work better for killing PARPiresistant senescent tumor cells; or (b) other senolytic drugs that target diverse senescence pathways could kill both PARPiresistant OV1369 and PARPi sensitive OV1946 cell lines. These two concepts were shown to work in in vitro models but no data was shown for these two ideas to work in in vivo animal tumor model.

Finally, other PARPi also induce senescence in ovarian cancer cells and reversible senescence was noted in breast cancer cells too. In short, the authors propose that results are universally applicable.

Response: We thank the reviewer for thoughtful comments, which have strongly helped us to clarify the message we would like to convey and to improve our manuscript. In the general comment above, we identify two specific points that the reviewer raises (underlined above in (a) “would work better for killing PARPi resistant senescent tumor cells” and (b) “These two concepts were shown to work in in vitro models but no data was shown for these two ideas to work in in vivo animal tumor model”).

To answer point (a), we have now explained more clearly in the text that the synergy between PARPi and ABT263 (or other senolytics) is not related to the PARPi resistance level, but rather to the induction of the senescence-like state, in the way that the synergy will only work in the context of a minimal induction of DNA damage and the senescence-like state by PARPi (senescence is related to the level of induction of DNA damage). This is illustrated in new Fig. 5 when multi-days senescence-inducing PARPi treatment is required to gain any synergistic effects. As expected higher *in vitro* concentrations of PARPi are necessary for the resistant cell lines to achieve comparable senescence

effects when compared to the sensitive cells. As suggested below in other comments by this reviewer, we have also improved throughout the manuscript our interpretations and comments regarding the differences observed between sensitive and resistant HGSOC cells, and we highlight that resistant cells harbor more senescence, that is, when we use relatively high concentrations of PARPi *in vitro* (Fig. 1). To answer point (b), as suggested by both reviewers, we supplement our original mouse pre-clinical experiments with PARPi-sensitive cell lines (old Fig. 4B, new Fig. 7B) by performing additional *in vivo* experiments using PARPi resistant OV90 HGSOC cells and MDA-231 breast cancer cells (new Fig. 7), which display an intermediate-resistance Olaparib phenotype (new Fig. 6A). Unfortunately, we could not perform *in vivo* evaluation of PARPi resistant OV1369(R2) cells because this cell line does not form tumors in mice (Letourneau et al. BMC Cancer 12, 379, 2012). While significant *in vivo* synergy was observed for all sensitive HGSOC cells tested and for the PARPi intermediate-resistant breast cancer cells MDA-MB-231 (new Fig. 7), no decrease in tumor growth was observed for the resistant OV90 cell line in any of the regimens tested (i.e. Olaparib or ABT263 alone, or the combination), albeit the use of higher concentrations of Olaparib than that used for the sensitive cell lines (new Fig.7A and Methods). We were unable to increase PARPi concentration further due to Animal Protection Ethics, and thus used the highest possible PARPi concentration *in vivo*. Importantly, as suggested by reviewer 2, we have confirmed that PARPi causes DNA damage in all xenograft models that responded to this treatment and found that indeed PARPi concentrations used *in vivo* were not sufficient to trigger any evidence of DNA damage in OV90 PARPi resistant tumors (new Fig. 7D-E). In summary, this supports our conclusion that BCL-family inhibitors or senolytics in general will have no synergistic effects with PARPi in the absence of a primary senescence induction by PARPi (as is more clearly illustrated in the new model presented in Fig. 8).

2) Are the claims and findings novel and will they be of interest to others in the community and wider field?

A previous paper Yokoyama et al. (2017) (ref 51 of the manuscript) has shown that PARPi (BMN673 or niraparib) with ABT263 synergistically cause more apoptosis in high grade ovarian cancer cell lines, such as OVCAR3, 8 and OV90. Since OV90 cell line is common in two studies and since end result is more cell death, there is an overlap of results between two studies. However, since Yokoyama group did not attempt to characterize the PARPi treated cells as senescent, the current manuscript could be accepted to have made a novel mechanistic explanation that senescence induced by PARPi treatment is the target for enhanced killing by senescence targeting drugs.

Moreover, Chang et al. 2016 (Nature Medicine, 22, 7983) had already identified that ABT263 kills senescent cells via apoptosis. Thus, while Yokoyama et al failed to connect this effect of ABT263, this manuscript effectively exploited this interpretation to make a clear and novel mechanistic argument of targeting senescent cells. Therefore, this finding would be of interest to the community and wider field.

Response: We are thankful for these positive and relevant comments. To improve on conveying the novelty of multistep targeting of senescent cancer cells we have more clearly integrated this concept in the updated model presented in new Figure 8. In addition, we have now added data to show that HGSOC have higher Bcl-XL expression than that of Bcl2 (TCGA data and our cell lines) (new Fig.4G), and that expression of Bcl-XL is enhanced after PARPi treatment (new Fig.4I, new Fig.6F-G), revealing that Bcl-XL is a particularly potent and specific senolytic target for HGSOC and TNBC.

3) Is the work convincing?

This is the weakest link in the paper. There is too much of data (2 PARPi resistant and 2 PARPi sensitive cell lines, variety of assays, and plethora of data which is grossly generalized across the board for all cell lines despite significant differences in results.

For example, 20% to 40% apoptosis is classed as similar level of death, or IL2 and IL8 differences among cell lines are glossed over. The differing degree of Knockdown for shp21 (50%) and shchk2 (80%) are treated at par KD level although extent of cell death varies between these two cell lines for each KD. The discordance between mRNA and protein levels for IL2 and 8 is glossed over. For many

other parameters, only mRNA is analyzed without a mention of what are the actual differences in protein levels in these cells. The lack of proper explanation for data variability among different figures erode the value of global data.

Response: To improve on conveying our interpretations of the rich dataset presented, we have substantially changed the text of our manuscript to better explain the discrepancies obtained, giving particular emphasis on the differences between the effects observed with resistant *versus* sensitive cell lines (pages 5, 8, 9, 10). As also suggested by reviewer #2, we conducted more in-depth analyses of p21 and Chk2 knockdown using two different shRNAs for each gene (two new shRNAs for p21 and one new for Chk2, see Methods) and performing new EdU labeling and clonogenic assays (new Figure 2). We now observed 75-80% mRNA reduction for both p21 and Chk2 proteins independent of the shRNA used. We then confirmed and completed the p21 and Chk2 knockdown results shown in our original manuscript and we hope that they are now more convincing for the reviewer. For the discrepancies between mRNA and protein levels of IL-6 and IL-8, it is possibly because the protein levels shown in our results are those secreted to the culture medium where proteins can accumulate, and the mRNA are intracellular reflecting a snapshot of the cellular steady-state, nevertheless the trends are clearly in the same directions (up- or down-regulated). Senescence phenotypes like the SASP are often defined by a multitude of markers (as we have scored) and the general trend of changes is usually more important to report rather than discussing individual factors, unless there is a focus on microenvironment, which is not the case in this manuscript. As suggested by the reviewer for Bcl2 and Bcl-XL, we have re-done some of our western blots and Q-PCR to improve clarity and we have now included additional TCGA data to confirm our observations in a large independent patient cohort (new Fig. 4, new Fig. 6, new Fig. S8, new Fig. S9).

4) Some big picture concepts relevant for manuscript have not been addressed:

A) If total death by combination of two drugs is due to PARPi induced direct death and ABT263 induced death of senescent cells (Fig. 4C), it should be expected that a sum of % of dead and senescent cells by PARPi alone as shown in Figs 1B and 1D respectively, should be reflected in total death caused by PARPi plus ABT263 (Fig. 4C).

However, this is not the case even at the lowest dose of PARPi in Fig 1B or D, which is expected to have been applied in Fig 4C. For example, for OV1369, 10% death and 20% senescence at low PARPi in Fig. 1, becomes 70% total death in Fig. 4C. There is no accounting for 40% additional death. For Ov1946, 45% death and 10% senescence at lowest PARPi in Fig.1 becomes 70% total death in Fig. 4. No efforts have been made to account for additional death (synergistic effect) of two drugs, which could be more in line with Yokoyama data explanations as more death due to more prodeath environment created by ABT263.

Response: Indeed, we think that the point made by the reviewer should be right, the combined amount of apoptotic + senescent cells in the PARPi alone treatment should be similar to the total amount of death observed in the PARPi+senolytic treatment (since the senolytic will convert senescent cells to dead cells). The discrepancy raised by the reviewer was caused by using the data from the low concentration of PARPi in Figure 1 to estimate the % of dead and senescent cells and applying the calculations to the combination therapy results reported in figure 4. For specific reasons, both figures do not use identical concentrations of drugs. Figure 1 is used to demonstrate PARPi dosage effects, in Figure 4 we have used only an optimized concentration of PARPi (IC₅₀) to allow discovery of synergistic effects when combined to ABT-263. In response to this comment, we have modified the text in the corresponding result section and the figure legend (new Figure 4) to make sure it is clear we are using PARPi IC₅₀ for combo treatments. In this context, we would like to point out that values presented in Figure 4C (new Fig. 4D) were obtained using Olaparib IC₅₀ concentrations specific for each cell line as shown in Figure 1, so if we follow the reasoning raised by the reviewer but perform the calculations using IC₅₀, the sum of dead and senescent cells in Figure 1 (OV1369(R2): ~10% dead + ~60% senescent; OV90: ~25% dead + ~40% senescent, OV4453: ~20% dead + ~40% senescent,

OV1946: ~40% dead + ~20% senescent) are very similar to the total cell death obtained in Figure 4C (new Fig. 4D) for the combinations (~70% cell death for all cell lines). So in addition to having, we hope, made our point more clearly, this mathematical exercise should reassure the reviewer that our results and interpretation are valid.

B) The reason for using PARPi resistant versus PARPi sensitive cell lines is not emerging in any of the data, as no effort has been made to correlate resistance with eventual response. An intuitive assumption could be that PARPi resistant cells would have more senescence (as an explanation of resistance) and therefore more profound effect of ABT263 co-admin with PARPi. But no such effect was anticipated, observed or explained.

Response: We apologize for not making our point clearer but the purpose of studying both sensitive and resistant PARPi cell lines was to cover a larger range of conditions that could be clinically relevant and to demonstrate that the senescent phenotype is a general process, but linked to appropriate concentrations where PARPi will generate primary DNA damage in the first place. The reviewer's intuition is correct, and indeed resistant cells show more senescence (*in vitro*) and combined to the other comments above from this reviewer we have been able to improve our understanding and explanations of the difference between sensitive and resistant cells. We also did our best to incorporate these concepts clearly in our proposed model (new Fig. 8) and new explanations at the end of the discussion (page 16).

C) The in vivo tumor data (Fig. 4F) shows combo PRRPi+ABT263 being effective only in PARPi sensitive cell lines (4453 and 1946) and no data is shown for any of the two PARPi resistance cell lines (1369 and 90). If we assume that for resistant cells, tumor data of combo treatment did not produce dramatic results, it would have been better to show animal data and then explain, as it was tried to be done in Fig 4G. On the other hand, the authors just add more data in vitro with other Senolytic agents (acting on different targets) and continue to make the argument with more in vitro data about universality of the combination effectiveness for PARPi resistant AND PARPi sensitive cell lines, while completely ignoring that fact that this did not work in vivo for PARPi resistant cell lines. The data series for other senolytic agents added after Fig 4 appears weak without support from in vivo tumor data. In summary, lack of strong support for the model in the in vitro data with mouse model is glossed over and not explained.

Response: This is an excellent point, and as mentioned above in response to point 1, we have now included the *in vivo* results for the resistant cell line OV90 (new Fig. 7), and we concluded that the Olaparib concentration achievable in mouse was not sufficient to induce primary DNA damage *in vivo* for OV90 tumors, thus explaining the lack of PARPi or synergistic effects (new Fig. 7). Unfortunately *in vivo* experiments cannot be performed with the other resistant cell line (OV1369(R2)) because it does not form tumors in mice. In place, as suggested by reviewer 2, we performed *in vivo* experiments with the breast cancer cell line MDA-MB-231 (intermediate-resistant) showing similar results as the ovarian cancer sensitive cell lines (new Fig. 7). Our results show that the *in vivo* concentration of Olaparib used for this breast cancer cell line was effective to induce DNA damage and senescence and thus make these tumors susceptible to senolytic synergy. This supports our improved model (new Figure 8) that propose PARPi must be able to induce DNA damage and senescence for any synergistic effect to be observed with senolytics. In essence, the reviewer was right in this comment, and allowed us to better explain the exact context in which synergistic effects between PARPi and senolytic drugs is expected.

5) What further evidence would be required to strengthen the conclusions?

Manuscript needs extensive revision with streamlined and consistent arguments strongly supported by in vitro and supporting in vivo data to make a strong case of the main argument.

Response: As explained in our response to the reviewers' comments above, we have substantially changed our manuscript, adding new data supporting our conclusions and making clear our main

observations. We have also better explained our proposed model in Figure 8 and in the Discussion section (Page 16).

6) Would the paper influence thinking in the field?

Yes, if the conclusions are strongly supported by consistent and clear data.

Response: Our efforts were concentrated on providing new data, and revising and changing our manuscript to make our conclusions based on solid findings. Therefore, we are confident that our paper has strong potential to influence thinking in the field.

7) Other questions and concerns?

Great potential for the therapeutic application of the work.

Response: We are thankful for these encouraging words, and we really think that pursuing avenues akin to those presented here could help improve survival in HGSOE (and possibly TNBC) patients.

8) Appropriateness and validity of statistical analyses, if applicable.

Could have shown more of raw data to support graphs and charts shown as the main data.

Response: We agree with the reviewer and have now added more representative raw data in some figures and as supplementary materials (new Fig. 4B-C, new Fig. 6H, new Fig. 7D, new Fig. S2A-B, new Fig. S4D, new Fig. S11A-E, new Fig. S12B-,D)

Reviewer #2 (Remarks to the Author):

1. Experiments shown in Figure 2 (Figure 2DH) testing for CHK2p21 mediated regulation of proliferation arrest, employ a single shRNA for depletion of p21 and Chk2. The specificity of the observed effects on cell number, % cell death, DNA damage and mitotic catastrophe would be best supported/confirmed with independent shRNA for each gene and is commonly expected/ required for publication in high quality journals.

Response: We agree with the reviewer's comments and have performed new experiments using two distinct shRNA against each p21 and Chk2 genes. Results are presented in the new Figure 2 (see also our response to reviewer #1 point 3).

2. Figure S4 is shown to establish the increase in expression of antiapoptotic genes BclXL and Bcl2. The mRNA expression levels support an increase in expression of both BclXL and Bcl2, but the WB shown for BclXL protein expression (Fig. S4C) shows little difference in 3/4 cell lines only OVC1946 shows clear/convincing increase in protein the parallels increase mRNA expression. The authors should comment on the effect of drugs in the absence of changes in protein expression.

Response: This is an important point. As explained in our response to reviewer #1 point 2), we have performed additional experiments to analyze the basal and induced levels of Bcl2 and Bcl-XL both at the mRNA and protein levels (new Fig. 4H-I, new Fig. 6F-G, new Fig. S8A, new Fig. S9C-D). In addition, we have interrogated the TCGA database to analyze the mRNA levels of these genes in HGSOE patients (new Figure 4G). Our results show that the Bcl-XL levels are higher than that of Bcl2, at both protein (our cell lines) and mRNA (our cell lines and the TCGA data) levels. In addition, BCL-XL levels are increased after Olaparib treatment, explaining increased efficacy of specific Bcl-XL inhibitors when compared to specific Bcl2 inhibitors (new Fig. 4I-J, new Fig. 6F-G, new Fig. S7, new Fig. S9C-D).

3. In regard to the ABT263 and Olaparib synergy, and possibly relevant to the point above, the concentration of ABT263 used seems high (micromolar concentrations). What are EC50s for ABT263 in HGSOE cells, what confirmation if any is there that at micromolar concentrations of ABT263 that

effects remain on target? The same question about off-target effects applies other senolytic agents used at high concentration.

Response: This is an important point, and suggests that the data in our original manuscript was not presented clearly. We originally presented the IC₅₀ and sensitivity levels for all our senolytic drugs in OV1369(R2) and OV1946 cells and the concentrations used in the synergy experiments were based on these results (Fig. 4G, Fig. S8A; now new Fig. S6C, new Fig. S10A). We have now modified the manuscript to clarify the experimental setting and to specifically point to the sensitivity data (page 8). The micromolar concentrations used are in accordance with previous publications (Chang et al, Nature Med. 22, 78, 2016; Yosef et al, Nature Comm. 7, 111190, 2016), however, as any targeted drug it is not possible to completely exclude an off-target effect. Nevertheless, the fact that we can use 3 different PARPi and multiple senolytics (including many Bcl2 family inhibitors) interchangeably strongly suggest that the on-target effects of either the PARPi or the Bcl2i drugs are generating the observed effects.

4. Experiments showing the in vivo effects (Figure 4) show xenografts for cell lines demonstrated to be sensitive to Olaparib in vitro. It would be of clinical interest/importance to know if the combination treatment show in vivo activity in the more PARPi resistant cells, as this may be a means to improve PARPi sensitivity to benefit patients who might otherwise not be expected to benefit from this class of drugs.

Response: Very important point. As explained in our response to reviewer #1 point1, and as suggested by this reviewer, we have now included pre-clinical results using the OV90 resistant cell line and the breast cancer cell line MDA-MB-231 (intermediate-resistant) (new Fig. 7). Unfortunately, we could not perform *in vivo* experiments with the OV1369(R2) cell line because it does not form xenograft tumors (Letourneau et al. BMC Cancer 12, 379, 2012). Please refer to the response to reviewer #1 for more technical details, but briefly, using intermediate-resistant and resistant cell lines has allowed to strengthen our model (improved Figure 8) based on the idea that PARPi must first induce DNA damage-senescence for any synergistic effects with senolytic drugs to occur. It appears that unfortunately, tumors that are very resistant to PARPi (i.e. PARPi has no effect to cause DNA damage), will not be made more sensitive to senolytics, as is the case for OV90 tumors. However, it remains possible that other DNA repair or cell cycle related synthetic lethal strategies could be used to render these cells more sensitive to PARPi in the first place, which could then make them amenable to senolytic synergies (perhaps through a tri-therapies strategy). We now discuss this at the end of the discussion section “We thus propose a two-step synergistic working model in which PARPi must first create DNA damage and senescence, a cellular state that can be targeted for senolysis (Fig.8). Although we present direct evidence for a two-step approach to exploit interconnected DNA repair-senescence synthetic lethalities, the model predicts that PARPi therapies (1st drug) can be enhanced at two levels, first via additional DNA damage- and cell cycle-related synthetic lethal approaches (2nd drug), which could create more DNA damage and senescent cells that are then targetable via senolysis (3rd drug).”

Additional information could be provided from the analysis of the tumors collected (Fig. 4F), such as tumor morphology and evidence of senescence, apoptosis and ontarget effects of olaparib and ABT263.

Response: We thank the reviewer for this important suggestion, we have performed immunohistochemistry for gH2AX (to validate DNA damage induced by PARPi) in all xenograft tumors collected 2 weeks after the beginning of the treatment. Because MDA-231 tumors grow very fast (compared to the HGSOc tumors that we have used) we were also able to collect fresh tumor tissue from the complete cohort 12 days after PARPi initiation allowing the evaluation of senescence biomarkers via RNA expression at an optimal time to measure senescence phenotypes, confirming evidence of senescence in this model (new Figure S13).

5. Evaluation of combination therapy in an MDAMB231 xenograft model would further support the in vitro results and the potential clinical application in breast cancer.

Response: We agree with the reviewer, and performed a full spectrum of *in vitro* and *in vivo* experiments using the breast cancer cell line MDA-MB231 to match data obtained with HGSOC (new Fig. 7). Importantly MDA-231 have an intermediate-resistant PARPi phenotype allowing us to use relatively PARPi-resistant cells *in vivo* to demonstrate synergistic effects. Results show a synergy between Olaparib and ABT263 both *in vitro* and in preclinical models.

Reviewers' comments:

Reviewer #1 (Remarks to the Author):

Major comments

1) The in vitro data described in pages 10 and 11 has significantly strengthened the revised manuscript. Reads better too.

2) I have one major and two minor comments on the presentation of the in vivo data (pre-clinical mouse model)

2A: Major comment: In vivo data issue for PARPi-resistant cells is still not resolved

The entire idea of putting both PARPi-sensitive and PARPi-resistant HGSOC in the manuscript should be reconsidered by the authors in light of the failure to generate in vivo data with both the PARPi resistant cells that could validate in vitro data. While one resistant cell line (OV1369) did not form the tumors in mice, another (OV90) did form tumors but they did not respond to therapy by PARPi +/- senolytic agents (Fig. 7B). Thus, this amounts to a technical issue of not having established a proper functional in vivo model for PARPi-resistant cells that can validate the in vitro data for these cells.

In addition, the explanations for OV90 tumor not responding to PARPi (+/- Senolytic agent) is not strongly supported by the data at two levels:

(i) Based on Fig 7C of 12th day tumor data, the conclusion is drawn that there is no effect of PARPi or combination with senescence inhibitor on OV90 tumor. However, the tumor growth curve in Fig 7B shows that PARPi (green line) was the only treatment that caused reduction in tumor size as compared to untreated control (black line), which was most evident on day 10.

This point needs to be addressed by providing statistical significance of difference at each time point of tumor growth data among different groups in Fig. 7B (for example by using Anova-2 factor with Bonferroni post test). If in tumor growth curves, there is effect of PARPi on day 10 but no effect of senolytic agent alone or combo of two agents, this will also need re-interpretation.

(ii) The lack of effect on OV90 tumor growth has been interpreted as PARPi not reaching the tumor in sufficient concentration to cause DNA damage (gH2AX at day 12, Fig 7D). This result has been taken as the basis that there would be no senescence response, which could be targeted by senolytic agent. However the single point analyses of DNA damage (gH2AX) at day 12 may not be the ideal method to claim that there was no DNA damage at any time in the 12 day protocol in which PARPi was given every day. It is entirely possible that DNA damage may have occurred in the early days of the protocol, but this was not verified.

Secondly, only convincing proof that senescence did not occur in OV90 model is to actually examine senescence parameters in OV90 tumors, as it was done for MDA-MB-231 tumors (Suppl data) but not done for the OV90 tumors.

Therefore, conclusion that PARPi somehow did not cause enough DNA damage and therefore could not cause enough senescence response to reduce OV90 tumor growth is not strongly supported by the data. It is also possible that OV90 in 2D-monolayer culture versus as xenograft 3D tumor may respond differently to PARPi.

Whatever may be the reason for this difference, presenting OV90 xenograft model in the manuscript with weak explanations actually reduces the value of rest of the manuscript-data that is very solid.

There are two suggestions for the authors: (a) Since in vitro data is convincing for all four ovarian cancer cells (PARPi-sensitive and resistant), as well as the TNBC cells (Moderate PARPi-resistant), the in vivo data of PARPi-resistant ovarian cells OV90 could be excluded, as it adds no value because the animal model is not suitable for the proposed experiments. In this case, manuscript may be reorganized as follows for presenting a coherent model validated in vitro and in vivo. The in vitro data of HGSOc cells should not be divided in PARPi-sensitive and resistant cells for in vitro studies is not necessary, just present in vitro data for four ovarian cell lines and one TNBC cell line. Then validate the in vitro data with in vivo cancer data for two ovarian cancer cell lines (PARPi-sensitive ones) and one TNBC cell line. Only new data that should be added is the analyses of senescence parameters in PARPi-sensitive tumor samples (similar to what is shown for TNBC tumor). This will make a strong paper with a clear message.

2B) Minor comment on in vivo data presentation:

Page 12: In fig 7A, the protocol for PARPi-sensitive cells runs up to 4 weeks (28 days), but tumor data shown in Fig 7B stops at day 21: why not add 28 day data too? The 28-day result would show even better difference for treated versus untreated groups. If 28 data is not to be included, then at a minimum, the cartoon of Fig 7A must be modified to 21 days.

3) Discussion: In the first paragraph, many general statements have been made about how PARPi causes DNA damage, senescence and hence makes PARPi-treated cells a better target for combo therapy with senolytic agent. All these statements are fully supported by in vitro data but not for part of in vivo data (as presented). Hence if the in vivo data is not presented differently, as suggested above, the text must be modified to emphasize the universal nature of this observation mainly in the in vitro studies.

Minor comments

1) Page 3: Full name of PARP1 is poly(ADP-ribose) polymerase (no space between poly and (ADP-ribose). Despite what we see more recently in some papers, poly(ADP-ribose) is chemically correct way to write its full name, which refers to polymer of ADP-ribose. As an example, we write other polymers without space: polyglutamine, polydA or poly(dI-dC), etc

2) page 3: paragraph 2:

“PARPi lead to DNA damage responses (DDR) that favor DNA repair, cell cycle checkpoints, and tissue remodeling (ref 7).”

This sentence needs to be modified for two reasons: (a) it is not attributable to reference 7 by Pommier: at least I could not find any statement in this paper to this effect about the role of PARPi; (b) The phrase creates a confusing logic: If DDR is caused by PARPi blocking the DNA repair, we can't continue the sentence by saying this DDR now favours DNA repair. Without PARPi, the rest of the sentence would have passed off as effect of DDR, but not with PARPi-DDR as starting point.

3) Page 4, 14 and 15: claims of “first” to show, do etc.

Are the claims for “First” allowed without proof that the authors have searched all databases? Let future reviewers make this claim in favour of the authors after extensive referral to the literature.

4) Page 6: Fig. S2B

PARPi-treated cells reveal formation of cells with DNA content beyond 4N, which could be integrated in the text with potential explanation overlapping with other results.

5) Page 7 onwards: There is a significant discordance between the sequence in which figures and their panels are referred to in the text versus arrangement of data figures (both for main and suppl figures). This creates an unnecessary strain in reading the manuscript and it would be problematic even for placement of figures in print version.

6) Page 8: line 2 (Fig. 3A-E)

what is "full" versus "clear" recovery? Explain better

7) Page 8: para 2, line 8: Typo: ...none (instead of no) or minimal effect..

8) Page 12: Typo:

were..

Overall, the manuscript presents a novel and therapeutically interesting approach for treatment of possibly many different cancers with PARPi and senolytic agent provided PARPi is able to cause DNA damage and senescence in these cancers.

Signed:

Girish Shah

Reviewer #2 (Remarks to the Author):

Manuscript#: NCOMMS-18-14775R

Corresponding Author: Francis Rodier

Title: Exploiting interconnected synthetic lethal interactions between PARP inhibition and cancer cell reversible senescence

This is an improved version of the previous submission. The authors provide substantial additional data and adequately address the comments and suggestions made by both reviewers. This manuscript merits publication.

The authors still summarize some of the data using generalizations that reviewer 1 previously noted as 'glossing over' some of the differences observed (examples below). The findings are well-supported by the data shown, and most thoughtful readers will understand that not all cell lines tested will behave exactly the same way and not discount the entirety of the findings based on some cell-line specific differences. More precise descriptions of the observations are recommended.

Comments:

1. Page 7 states “we verified the Chk2 (CHK2) expression by Q-PCR in cells treated with Olaparib for 6 days and observed significant upregulation in all four cell lines (Fig.2G)”, but the data show the effect was significant in three of four cell lines.
2. The shRNA depletion experiments now employ two independent shRNA constructs for each gene and mRNA depletion for both is shown, but depletion of protein is not shown. Presumably western blots were run to confirm knockdown of protein as well as mRNA. These the corresponding images should be shown.
3. Page 7 states “Depleting either p21 or Chk2 prevented the proliferation arrest of Olaparib-treated HGSOC cells” – the effect on proliferation is incomplete, particularly for CHK2 in OV1369 cells, thus this should read “partially prevents”. Similarly, it is stated that depletion of p21 or CHK2 redirects olaparib treated cells toward cell death, yet there are no differences in ratio of cell number or % cell death in OV1369 cells with either shp21.4 or shp21.6 compared to control.
4. Page 8 – “To highlight potentially additive or synergistic effects” should read additive or synergistic.
5. Page 10 – the statement “Indeed, levels of γ H2AX and 53BP1 nuclear foci (Fig.S9F-H) or secreted IL-6 and IL-8 (Fig.S9I-J) after release reverted to the same levels as matched controls; however, this effect was not as pronounced in sensitive cells,” is not consistent with results observed in OV4453 cells – not simply not as pronounced, but reversed.

March 4th, 2019

Exploiting interconnected synthetic lethal interactions between PARP inhibition and cancer cell reversible senescence

Hubert Fleury*, Nicolas Malaquin*, Véronique Tu, Sophie Gilbert, Aurélie Martinez, Marc-Alexandre Olivier, Alexandre Sauriol, Laudine Communal, Kim Leclerc-Desaulniers, Euridice Carmona, Diane Provencher, Anne-Marie Mes-Masson# and Francis Rodier# (* These authors contributed equally; # corresponding authors)

Point by point response:

Black= reviewer's text

Blue= author's response

Reviewer #1 (Remarks to the Author):

Major comments

1) The in vitro data described in pages 10 and 11 has significantly strengthened the revised manuscript. Reads better too.

We are thankful for this positive comment.

2) I have one major and two minor comments on the presentation of the in vivo data (preclinical mouse model)

2A: Major comment:

In vivo data issue for PARPi resistant cells is still not resolved. The entire idea of putting both PARPi sensitive and PARPi resistant HGSOc in the manuscript should be reconsidered by the authors in light of the failure to generate in vivo data with both the PARPi resistant cells that could validate in vitro data. While one resistant cell line (OV1369) did not form the tumors in mice, another (OV90) did form tumors but they did not respond to therapy by PARPi +/-senolytic agents (Fig. 7B). Thus, this amounts to a technical issue of not having established a proper functional in vivo model for PARPi resistant cells that can validate the in vitro data for these cells. In addition, the explanations for OV90 tumor not responding to PARPi (+/Senolytic agent) is not strongly supported by the data at two levels: **(i)** Based on Fig 7C of 12th day tumor data, the conclusion is drawn that there is no effect of PARPi or combination with senescence inhibitor on OV90 tumor. However, the tumor growth curve in Fig 7B shows that PARPi (green line) was the only treatment that caused reduction in tumor size as compared to untreated control (black line), which was most evident on day 10. This point needs to be addressed by providing statistical significance of difference at each time point of tumor growth data among different groups in Fig. 7B (for example by using Anova2 factor with Bonferroni post test). If in tumor growth curves, there is effect of

PARPi on day 10 but no effect of senolytic agent alone or combo of two agents, this will also need reinterpretation.

The reviewer is suggesting a strategy to re-analyze the presented OV90 tumor data in order to show that the PARPi alone treatment may have a statistically significant effect specifically at 10 days' time into the treatment. We have verified this possibility using the suggested statistical strategy (2-way Anova with Bonferroni post-test), and the effect of PARPi treatment versus control is not significant in OV90 at day 10 or at any time points (see attached Editorial table 1). In fact, there are no other time-points with notable significant differences that could change the overview presented in figure 7C. Of note, these results reinforce the suggestion below (see section ii) by this reviewer, and we have now removed all OV90 pre-clinical data from the manuscript, which also addresses any over interpretation of the data. Thus, the current version of the manuscript presents the *in vivo* models (OV4453/OV1946 ovarian cell lines and MDA-MB-231 breast cancer cell line) that all have proper statistics demonstrated for the experimental endpoint presented in Figure 7C. Although we agree with the general idea of presenting all relevant statistics, given the obvious separation between the curves observed for the remaining models in Figure 7B, and the statistics for the relevant time-point presented in figure 7C, we decided not to add the 2-way ANOVA tests with Bonferroni post-corrections between all possible options because it cannot be usefully presented in the Figure 7B graph. However, we have attached the full statistical reports for the editor and reviewers to examine to address this point (see attached Editorial tables 2-4).

(ii) The lack of effect on OV90 tumor growth has been interpreted as PARPi not reaching the tumor in sufficient concentration to cause DNA damage (gH2AX at day 12, Fig 7D). This result has been taken as the basis that there would be no senescence response, which could be targeted by senolytic agent. However, the single point analyses of DNA damage (gH2AX) at day 12 may not be the ideal method to claim that there was no DNA damage at any time in the 12 days protocol in which PARPi was given every day. It is entirely possible that DNA damage may have occurred in the early days of the protocol, but this was not verified. Secondly, only convincing proof that senescence did not occur in OV90 model is to actually examine senescence parameters in OV90 tumors, as it was done for MDAMB231 tumors (Suppl data) but not done for the OV90 tumors. Therefore, conclusion that PARPi somehow did not cause enough DNA damage and therefore could not cause enough senescence response to reduce OV90 tumor growth is not strongly supported by the data. It is also possible that OV90 in 2Dmonolayer culture versus as xenograft 3D tumor may respond differently to PARPi. Whatever may be the reason for this difference, presenting OV90 xenograft model in the manuscript with weak explanations actually reduces the value of rest of the manuscript data that is very solid.

The reviewer makes an important point, there are many possible explanations as to why a cell line would be treatment resistant *in vivo*, beyond its direct response to PARPi via the generation of DNA damage (for example, the drug may not reach the tumor effectively as opposed to delivery in 2D cultures, or the cells perhaps simply alter their biology). Because of this, we now realize that using the term PARPi-resistant versus PARPi-sensitive was perhaps not the best choice, as we do not want to make any confusing assumptions between the amount of PARPi required to cause DNA damage in a cell versus a tumor complex "resistance to treatment" *in vivo*. Based on the reviewer conceptual suggestion, we have reformulated our nomenclature for the cell lines used, which is still based on the required PARPi concentrations to induce DNA damage in 2D cultures

but now termed: Olaparib IC₅₀-low (OV4453 and OV1946), IC₅₀-intermediate (MDA-MB-231) and IC₅₀-high (OV90 and TOV1369). We have also limited the use of these terms, but they remain useful to support some important comments in the result section that were suggested in the previous review rounds regarding more detailed description of the data obtained with the 4 ovarian cancer cell lines, particularly when the data trends differ between each cell line. We hope this can avoid any confusion derived from using the term “resistant”, which has been removed throughout except in the discussion section where a point is specifically made with regard to patient options to circumvent treatment resistance. With this change in mind, we also agree with the reviewer that presenting *in vivo* data for only one PARPi-high (or previously termed PARPi-resistant) cell line OV90 is not useful as this data may not translate to other PARPi-high cell lines for many reasons including the multiple mechanisms that could result in treatment resistance *in vivo* other than the simple sensitivity to PARPi or the achievable doses. We also agree that this data is not required to support the otherwise reproducible data obtained *in vivo* with the other 2 ovarian cancer cell lines and for the breast cancer cell line. In summary, as suggested below by the reviewer, we have opted to streamline the manuscript via a modification of our PARPi sensitivity nomenclature to avoid the term resistance to describe our experiments and by removing the *in vivo* data for OV90. We think this makes the manuscript easier to read and still support the concept that we want to present, the idea that PARPis induce a reversible p53-independent senescence-like state that can be combined to senescence-targeting drugs to potentiate treatment effects.

There are two suggestions for the authors: (a) Since *in vitro* data is convincing for all four ovarian cancer cells (PARPi sensitive and resistant), as well as the TNBC cells (Moderate PARPi resistant), the *in vivo* data of PARPi resistant **ovarian cells OV90 could be excluded**, as it adds no value because the animal model is not suitable for the proposed experiments. In this case, manuscript may be reorganized as follows for presenting a coherent model validated *in vitro* and *in vivo*. The *in vitro* data of HGSOc cells should not be divided in PARPi sensitive and resistant cells for *in vitro* studies is not necessary, just present *in vitro* data for four ovarian cell lines and one TNBC cell line. Then validate the *in vitro* data with *in vivo* cancer data for two ovarian cancer cell lines (PARPi sensitive ones) and one TNBC cell line. **Only new data that should be added is the analyses of senescence parameters in PARPi sensitive tumor samples (similar to what is shown for TNBC tumor). This will make a strong paper with a clear message.**

We thank the reviewer for these suggestions. As explained above we accept and followed the suggestion to re-organize the manuscript to remove the terminology related to treatment “resistance” and we now refer simply to PARPi sensitivity in 2D culture models to classify the cell lines used. We also agree with the idea that the OV90 *in vivo* data can be removed from the manuscript without affecting the data gathered using ovarian (OV4453 and OV1946) breast (MDA-MB-231) cell lines.

Regarding the suggestion of adding additional *in vivo* senescence data, we indeed understand the rationale behind this request. But as explained below, we are limited by technical difficulties. As noted in our previous response, the pre-clinical MDA-231 model has provided us with technical advantages that are unfortunately not available in the ovarian cancer models. We would like to quote text from our previous response and comment further on the subject:

Text from previous response: “Because MDA-231 tumors grow very fast (compared to the HGSOc tumors that we have used) we were also able to collect fresh tumor tissue from the complete cohort 12 days after PARPi initiation allowing the evaluation of senescence biomarkers

via RNA expression at an optimal time to measure senescence phenotypes, confirming evidence of senescence in this model (new Figure S13).”).

Further comments: To add to this explanation, as is now clear, the OV90 model does not respond to PARPi *in vivo* for reasons difficult to establish (treatment resistance), and OV1369 do not form xenograft preventing the use of this model. While the OV4453 and OV1946 models allow us to develop pre-clinical cancer models, we would like to emphasize some limitations for evaluating senescence hallmarks *in vivo* with these models related to the slow tumor growth rates. OV4453 and OV1946 tumors grow slowly with delays of 2-3 months before we can begin an experiment. Accordingly, the experimental timeline to highlight differences in ovarian tumor growth rate also has to extend to 20-30 days from treatment initiation (as presented in Figure 7A-C; or about 3-4 months from cancer cell injection). This is unlike MDA-MB-231, which have fast tumor growth rates requiring short experimental treatment time to reach statistically significant differences in tumor growth rate. Importantly, MDA-MB-231 allow us to collect both growth data and tumor tissues within 10-12 days of treatment initiation. This characteristic is essential and it has allowed us to effectively quantify relevant senescence hallmarks at an optimal moment in time after treatment initiation when we expect a good senescence response (MDA-MB-231 model Figure S13). Unfortunately, in the OV4453 and OV1946 protocols, mice are sacrificed either at tumor volume endpoint or 28 days post-treatment initiation. At both of those endpoints, individual tumors often show a phenotype of slowed but continued growth, suggesting some cells are beginning to develop resistance *in vivo*.

Nevertheless, in an attempt to answer the above suggestion, we went ahead and extracted RNA from paraffin-embedded formalin fixed tumors collected at the 28d endpoint for the OV4453 and OV1946 experiments presented in figure 7A-C. Quality of the RNA was validated from good quantitative reverse-transcriptase PCR cycle-thresholds (qRT-PCR CT) obtained for two control transcripts that correlated well with each other (18s Ribosomal RNA and Tata-binding protein (TBP) Editorial Figure 1). Overall, we observed no significant differences in any of the senescence hallmarks we tested (Bcl-XL, p21, IL-8, IL-6) suggesting that at this moment in time senescence was absent in the tumors (Editorial Figure 2A-B). Unfortunately, this is not unexpected as unlike MDA-MB-231 these cohorts were not designed to detect senescence *in vivo*. In order to properly test senescence in these models at the best optimal time point to measure this response, we would have to initiate again full cohorts of OV4453 and OV1946 with the intent of sacrificing the mice at days 10-12. In fact, because the experiment itself would need to be validated for successful impact on tumor growth (in case of a negative result for senescence hallmarks at the selected time point), the cohorts would have to be twice as large as in our reported results. This would require the use of a large number of additional experimental animals to validate data that is already clear using the MDA-MB-231 model. However, we consider that this experiment is not crucial to further support the solid data that we have presented for the cell culture and pre-clinical models and we hope that for the reasons presented above the *in vivo* senescence data presented for the MDA-MB-231 will be sufficient to support the novel conclusions of our manuscript.

2B: Minor comment on in vivo data presentation:

Page 12:

In fig 7A, the protocol for PARPi sensitive cells runs up to 4 weeks (28 days), but tumor data shown in Fig 7B stops at day 21: why not add 28 days data too? The 28 days result would show

even better difference for treated versus untreated groups. If 28 data is not to be included, then at a minimum, the cartoon of Fig 7A must be modified to 21 days.

We thank the reviewer for this observation that will increase the concordance between models and data. We have changed the cartoon for a protocol in ovarian cancer cells running up to 21 days (instead of 28 days).

3) Discussion:

In the first paragraph, many general statements have been made about how PARPi causes DNA damage, senescence and hence makes PARPi treated cells a better target for combo therapy with senolytic agent. All these statements are fully supported by *in vitro* data but not for part of *in vivo* data (as presented). Hence if the *in vivo* data is not presented differently, as suggested above, the text must be modified to emphasize the universal nature of this observation mainly in the *in vitro* studies.

We have re-organized the text and the presentation of the *in vivo* data including removal of the negative OV90 data leading to some changes in the discussion. Overall, as suggested by the reviewer, to increase the concordance between our cell culture and pre-clinical model data, we have now made changes to refer specifically to cell culture models or to pre-clinical models when appropriate in the discussion.

Minor comments

1) Page 3:

Full name of PARP1 is poly(ADPribose) polymerase (no space between poly and (ADPribose). Despite what we see more recently in some papers, poly(ADPribose) is chemically correct way to write its full name, which refers to polymer of ADPribose. As an example, we write other polymers without space: polyglutamine, polydA or poly(dIdC), etc

We agree with the reviewer's comments and have remove space between poly and (ADPribose) in page 2, 3.

2) page 3: paragraph 2:

“PARPi lead to DNA damage responses (DDR) that favor DNA repair, cell cycle checkpoints, and tissue remodeling (ref 7).” This sentence needs to be modified for two reasons: (a) it is not attributable to reference 7 by Pommier: at least I could not find any statement in this paper to this effect about the role of PARPi; (b) The phrase creates a confusing logic: If DDR is caused by PARPi blocking the DNA repair, we can't continue the sentence by saying this DDR now favours DNA repair. Without PARPi, the rest of the sentence would have passed off as effect of DDR, but not with PARPi DDR as starting point.

We apologize for this confusing sentence. Our intention was to emphasize that PARPi leads to DNA damage accumulation (stated in ref 7 and also shown in Figure 1G and supplementary figure S1 G of our manuscript by increased numbers of gammaH2AX foci), which will subsequently trigger DDR. The sentence on page 3 (paragraph 2) was changed to better explain this point and to remove the ambiguity of favoring DNA repair. The new sentence reads: “PARPi induce DNA

damage accumulation⁷, leading to DNA damage responses (DDR) that favors cell cycle checkpoints and tissue remodeling.”

3) Page 4, 14 and 15:

claims of “first” to show, do etc. Are the claims for “First” allowed without proof that the authors have searched all databases? Let future reviewers make this claim in favour of the authors after extensive referral to the literature.

We agreed with the reviewer’s comments and removed these inappropriate claims from pages 4, 14 and 15.

4) Page 6: Fig. S2B

PARPi treated cells reveal formation of cells with DNA content beyond 4N, which could be integrated in the text with potential explanation overlapping with other results.

We thank the reviewer for this suggestion. As proposed, we have integrated this observation on page 6 as follow: “This was confirmed by a cell cycle analysis at 6 days post-treatment showing the accumulation at the G2/M phases of the cell cycle and appearance of cell populations presenting DNA content beyond 4N (Fig.1J, Fig.S2B), consistent with described effects of Olaparib on the cell cycle³⁰.”

5) Page 7 onwards:

There is a significant discordance between the sequence in which figures and their panels are referred to in the text versus arrangement of data figures (both for main and suppl figures). This creates an unnecessary strain in reading the manuscript and it would be problematic even for placement of figures in print version.

We understand that the discordance in the sequence in which figures and their panels are referred to in the text versus arrangement of data figures relates to Figures 2 and S4, where we described the effects of p21 and Chk2 knockdowns. Therefore, to improve clarity the order of the panels in these two figures were changed to match the sequence that they appear in the text (new Figure 2).

6) Page 8: line 2 (Fig. 3AE)

what is “full” versus “clear” recovery? Explain better

We agree that this sentence was not properly formulated. To be more specific in the new sentence we now re-use exactly the terms from the previous sentence that introduced “early senescence” versus “senescence” and we added the number of days. We have removed “full” versus “clear” as these terms are qualitative and replaced them with specific references to the data for cell number and DNA synthesis instead of amalgamating the results.

7) Page 8: para 2, line 8:

Typo: ...none (instead of no) or minimal effect..

We thank the reviewer for finding this typo error and corrected “no” by “none”.

8) Page 12:

Typo: were..

We thank the reviewer for finding this typo error and corrected “was” by “were.

Reviewer #2 (Remarks to the Author):

Comments:

1. Page 7

states “we verified the Chk2 (CHK2) expression by QPCR in cells treated with Olaparib for 6 days and observed significant upregulation in all four cell lines (Fig.2G)”, but the data show the effect was significant in three of four cell lines.

We thank the reviewer for nothing this detail. As proposed, we have modified in page 6 the sentence “in all the four cell lines” by “in three of the four cell lines”.

2. The shRNA depletion experiments now employ two independent shRNA constructs for eachgene and mRNA depletion for both is shown, but depletion of protein is not shown. Presumably western blots were run to confirm knockdown of protein as well as mRNA. These the corresponding images should be shown.

We agree with the reviewer’s comments and added the corresponding Western Blot images to Figure S4A .

3. Page 7

states “Depleting either p21 or Chk2 prevented the proliferation arrest of Olaparib treated HGSOC cells” the effect on proliferation is incomplete, particularly for CHK2 in OV1369 cells, thus this should read “partially prevents”. Similarly, it is stated that depletion of p21 or CHK2 redirects olaparib treated cells toward cell death, yet there are no differences in ratio of cell number or % cell death in OV1369 cells with either shp21.4 or shp21.6 compared to control.

We agree with the reviewer’s comments and modified sentence on page 7 that now reads: “Depleting either p21 or Chk2 partially prevented...”

Regarding our statement that “depletion of p21 or CHK2 redirects olaparib treated cells toward cell death”, we would like to point out that significant decrease in colony formation was observed for both cell lines when using either shp21 or shChk2 (Figure 2,L). However, it is true that no differences in ratio of cell number or % cell death was found for the OV1369(R2) cells with either shp21.4 or shp21.6 when compared to control. We therefore modified the sentence on page 7 accordingly, and it reads as follow: “Finally depleting either p21 or Chk2 redirected Olaparib-treated cells from senescence towards cell death, as observed by significantly decreased colony formation (Fig.2K,L). In addition, except for OV1369(R2) treated by shp21, significantly lower cell number ratio and higher cell death was observed (Fig.S5A-C).”

4. Page 8

“To highlight potentially additive of synergistic effects” should read additive or synergistic.

We thank the reviewer for finding this typo error and “of” was replaced by “or”.

5. Page 10

the statement” Indeed, levels of γ H2AX and 53BP1 nuclear foci (Fig.S9FH) or secreted IL6 and IL8 (Fig.S9IJ) after release reverted to the same levels as matched controls; however, this effect was not as pronounced in sensitive cells,” is not consistent with results observed in OV4453 cells – not simply not as pronounced but reversed.

We agree with the reviewer’s comments and modified sentence on page 10 to highlight this observation. The sentences now read: “Indeed, levels of γ H2AX and 53BP1 nuclear foci (Fig.S9F-H) or secreted IL-6 and IL-8 (Fig.S9I-J) after release reverted to the same levels as matched controls for the OV1369(R2) and OV90 cell lines. However, this effect was less pronounced for the OV1946 cells and even sometimes reversed for OV4453, consistent with the slower growth recovery curves for these cells (Fig.3A-E).”

REVIEWERS' COMMENTS:

Reviewer #1 (Remarks to the Author):

Authors have addressed all the concerns raised by me. The manuscript reads very well and presents a strong argument for improving PARPi therapy for more than one cancers via timing of supplementary targeting of p53 independent senescence. It will make a stronger argument for conducting clinical trials to validate these approach.

Girish Shah